# Searching for the cellular underpinnings of the selective vulnerability to tauopathic insults in Alzheimer's disease
Justin Torok [1], Pedro D. Maia [2], Chaitali Anand [3] & Ashish Raj [1] ✉

Neurodegenerative diseases such as Alzheimer's disease exhibit pathological changes in the brain that proceed in a stereotyped and regionally specific fashion. However, the cellular underpinnings of regional vulnerability are poorly understood, in part because whole-brain maps of a comprehensive collection of cell types have been inaccessible. Here, we deployed a recent cell-type mapping pipeline, Matrix Inversion and Subset Selection (MISS), to determine the brain-wide distributions of pan-hippocampal and neocortical cells in the mouse, and then used these maps to identify general principles of cell-type-based selective vulnerability in PS19 mouse models. We found that hippocampal glutamatergic neurons as a whole were significantly positively associated with regional tau deposition, suggesting vulnerability, while cortical glutamatergic and GABAergic neurons were negatively associated. We also identified oligodendrocytes as the single-most strongly negatively associated cell type. Further, cell-type distributions were more predictive of end-time-point tau pathology than AD-risk-gene expression. Using gene ontology analysis, we found that the genes that are directly correlated to tau pathology are functionally distinct from those that constitutively embody the vulnerable cells. In short, we have elucidated cell-type correlates of tau deposition across mouse models of tauopathy, advancing our understanding of selective cellular vulnerability at a whole-brain level.

A hallmark of all neurodegenerative diseases (NDDs), including Alzheimer's disease (AD), frontotemporal lobar dementia (FTLD), and other tauopathies, is the display of spatially heterogeneous and stereotyped distributions of brain pathology. In AD, accumulations of misfolded microtuble-associated protein tau ($\tau$), which are strongly associated with and precede neurodegeneration, are first observed in the locus coeruleus (LC) and then entorhinal cortex (EC), followed by orderly spread into limbic and temporal areas, the basal forebrain, and finally to neocortical association areas[1]. This propensity of AD tauopathy to canonically affect some regions but not others is referred to as *selective regional vulnerability*. Understanding the underlying factors that govern regional vulnerability is not only an important scientific goal, but will also aid in identifying targets for intervention, an effort that has gained urgency with the prevalence of NDDs on the rise and few therapeutic options for slowing their progression.

It has long been thought that regional vulnerability must be a consequence of the molecular composition of certain regions, which is in turn governed by upstream genes that are involved in disease pathology[2]. There

exists a notable dissociation between where AD risk genes are normally located in the brain and downstream pathology, an observation that has been called one of the key mysteries in the field of neurodegenerative diseases[3,4]. Taking a broader perspective on the interactions between gene expression and the development of AD-associated pathology may be required; for instance, Sepulcre, et al. examined propagation patterns of $\tau$ and amyloid-$\beta$ (A$\beta$) and found associations for hundreds of genes previously unknown to be associated with AD, utilizing transcriptomic data from the Allen Human Brain Atlas[5,6]. Furthermore, the dissociation between AD risk gene expression and selective regional vulnerability may be partly due to the fact that certain cell types, especially subtypes of glutamatergic neurons, in affected regions harbor significantly more $\tau$ inclusions and degenerate at a faster rate than others.

Great progress in exploring this phenomenon, known as *selective neuronal vulnerability*, has been facilitated by rapid advancements in single-cell sequencing technologies, allowing for the identification of key subpopulations of vulnerable neurons[7–11], which can be further separated into

[1]University of CAlifornia, San Francisco, Department of Radiology, San Francisco, CA, 94143, USA. [2]University of Texas at Arlington, Department of Mathematics, Arlington, TX, 76019, USA. [3]University of CAlifornia, San Francisco, Institute for Neurodegenerative Diseases, San Francisco, CA, 94143, USA. ✉e-mail: ashish.raj@ucsf.edu

*primary* (i.e., affecting a region's propensity to endogenously produce pathological $\tau$) and *secondary* (i.e., affecting how likely a region will be invaded by existing $\tau$ pathology elsewhere)[12]. For instance, emerging evidence suggests that neurons in the noradrenergic nuclei might be instrumental in initiating tau pathology[13]. Within the EC, $\tau$-inclusions are predominantly observed in *RORB*-expressing excitatory neurons, while other subpopulations appear relatively unaffected[11]. In the hippocampus, $\gamma$-aminobutyric-acid-secreting (GABAergic) inhibitory neurons, especially the *Pvalb+* and *Sst+* interneurons, tend to colocalize with $\tau$[14]. More generally, myriad of factors, from cytoarchitecture and potential exposure to external pathogens[15], to morphological attributes[16–21], are postulated to influence a cell's vulnerability or resilience to AD's pathological processes.

The role of non-neuronal cells in the disease process is also gaining wider attention. Oligodendrocytes, for instance, have been observed to sequester extracellular $\tau$, potentially leading to its internal amplification and subsequent dissemination[22,23]. Microglia and the broader process of neuroinflammation are widely acknowledged as pivotal mediators of both $\tau$ and amyloid-$\beta$ (A$\beta$) pathophysiology[24–32]. Astrogliosis, characterized by the proliferation of astrocytes in response to cellular damage, is also a significant focus in AD research[33–35].

This study seeks to gain insights into the foundations of *cell-type-based selective vulnerability* (SV-C) and *resilience* (SR-C) to $\tau$ at a whole-brain level from a statistical perspective. We seek not only to identify selectively vulnerable or resilient cell types, but also the general organizing principles that may emerge thereby. We are especially interested in understanding whether SV-C and SR-C are secondary to or independent of genetically conferred selective vulnerability and resilience (herein referred to as SV-G and SR-G, respectively).

Most current evidence on SV-C and SR-C has understandably come from descriptive, hypothesis-driven mechanistic experimental bench or animal studies, and limited to selected brain structures or selected cell types. These studies, based on a chosen transgenic animal model or specific $\tau$ conformational strain—that is, distinct molecular species of $\tau$ that result from aberrant post-translational modifications and misfolding[36]—are difficult to generalize. For instance, the roles of support cells such as microglia and astrocytes are complex, and whether they are helpful or harmful in tauopathy remains inconclusive and may be context-dependent[37,38]. We propose that a fuller assessment of broad phenomena like selective vulnerability would benefit from a principled integration of a diversity of data sources. We therefore employed a meta-analytical approach that integrated various histological $\tau$ data sourced from 5 distinct studies and encompassing a total of 12 experimental conditions, all of which used PS19 mouse models of tauopathy[39–43].

While great strides have been made in spatial transcriptomics in recent years[44–48], a comprehensive, whole-brain spatial atlas of cell types has not yet been developed, precluding the analyses required to examine SV-C and SR-C at a holistic level. However, quantitative cell-type maps at the regional level can be obtained using computational methods for performing spatial deconvolution of bulk spatial transcriptomic data[49–51]. Here, we utilized the recently described Matrix Inversion and Subset Selection (MISS) algorithm[49] to infer the distributions of 42 neuronal and non-neuronal cell types, utilizing the recent single-cell atlas developed by the Allen Institute for Brain Science (AIBS), which profiled approximately 1.3 million cells sampled across the hippocampal formation and neocortex[52]. This computational technique enabled us to significantly expand the number of covered cell types and their corresponding spatial coverage beyond those for cell types obtained by current in situ sequencing methods. These inferred cell-type densities could then be compared with regional distributions of $\tau$ quantified in mouse models of tauopathy, allowing us to probe questions of how the presence of cell types in a given region confers vulnerability or resilience to the later development of $\tau$ pathology; that is, *innate* SV-C and SV-R.

Our analysis revealed that, among the types of neurons present, hippocampal glutamatergic neurons were the only class that was consistently positively associated with $\tau$ pathology. Conversely, oligodendrocytes emerged as the cell type most negatively associated with pathology, suggesting a potential role in conferring resilience to $\tau$ at a whole-brain level. Furthermore, we found that combinations of cell types generally yielded more predictive multivariate models of $\tau$ pathology when compared with equivalent models utilizing the expression of 24 key AD risk genes[53,54]. A further exploration of the transcriptomic underpinnings of SV-C and SR-C revealed that there were surprising differences between the gene sets most strongly correlated with $\tau$ and those that were differentially expressed in vulnerable and resilient cell types. These differences were evident both in terms of the identified genes and their functional enrichment at the biological process level, shedding light on the distinct roles played by genes and cells, a finding that can help explain the dissociation between upstream genes and downstream pathology common to many neurodegenerative diseases[3,4]. Overall, our integrative modeling approach represents the broadest exploration of whole-brain selective vulnerability to date and offers a complementary framework to interpret the extensive pathology and -omics data generated by bench scientists, yielding generalized insights into selective vulnerability in tauopathies.

## Results

### Inferring new brain-wide maps for neuronal and non-neuronal cell types

We used the recently published Matrix Inversion and Subset Selection (MISS) algorithm to infer whole-brain cell-type distributions[49]. In brief, MISS determines distributions of cell types, as characterized by their single-cell RNA sequencing (scRNAseq) gene expression signatures, by performing a deconvolution on spatial gene expression data. We have previously demonstrated that MISS is an accurate, efficient, and robust tool for determining the whole-brain distributions of mouse neural cell types from two independently sampled scRNAseq datasets[55–57], where we used the coronal Allen Gene Expression Atlas (AGEA) as our source of spatially resolved gene expression data[49,58].

In the present study, we again leveraged the AGEA, but instead mapped the cell types recently characterized by Yao, et al.[52] (Fig. 1A). We chose to produce new maps using this scRNAseq dataset (herein referred to as the "Yao, et al. dataset") rather than rely on previously published cell-type distributions for two reasons: 1) Yao, et al. comprehensively sampled the neocortex and hippocampal formation, sequencing approximately 1.3 million individual cells with excellent read depth[52]; and 2) capturing cell-type-specific vulnerability to tauopathy requires robust and specific delineation of cortical and hippocampal cell types. Following our established procedure, we first obtained an informative 1300-gene subset optimal for the spatial deconvolution problem, and then solved the nonnegative matrix inversion problem voxel-by-voxel (Fig. 1B; see also Methods). The residual error at both the regional and voxel-wise levels was not greater outside of the neocortex and hippocampus, demonstrating that these whole-brain maps did not contain systematic errors outside of the regions where the mapped cell types were sampled (Fig. 1B). We also validated our results using published distributions of *Pvalb+*, *Sst+*, and *Vip+* interneurons in the neocortex and show excellent agreement with our inferred maps (Fig. 1B; Fig. S1).

In all, we mapped 36 neuronal and 6 non-neuronal cell types at the 200 $\mu$m resolution of the AGEA (Fig. 1C). We further classified these cell types into four major classes: cortical glutamatergic neurons, hippocampal glutamatergic neurons, GABAergic neurons, and non-neuronal cells (Tables S1 and S2 for complete descriptions of these cell types). The spatial structure of the resulting cell-type distributions was rich and diverse (Fig. 1D), and in fact cell types and cell-type classes (e.g., non-neuronal cells) have more varied regional densities than gene expression profiles (Figs. S2–S6). We emphasize that both the scRNAseq and spatial gene expression data were taken from healthy, adult, wild-type mice, and thus the MISS-inferred cell-type distributions represent *baseline* cell-type densities.

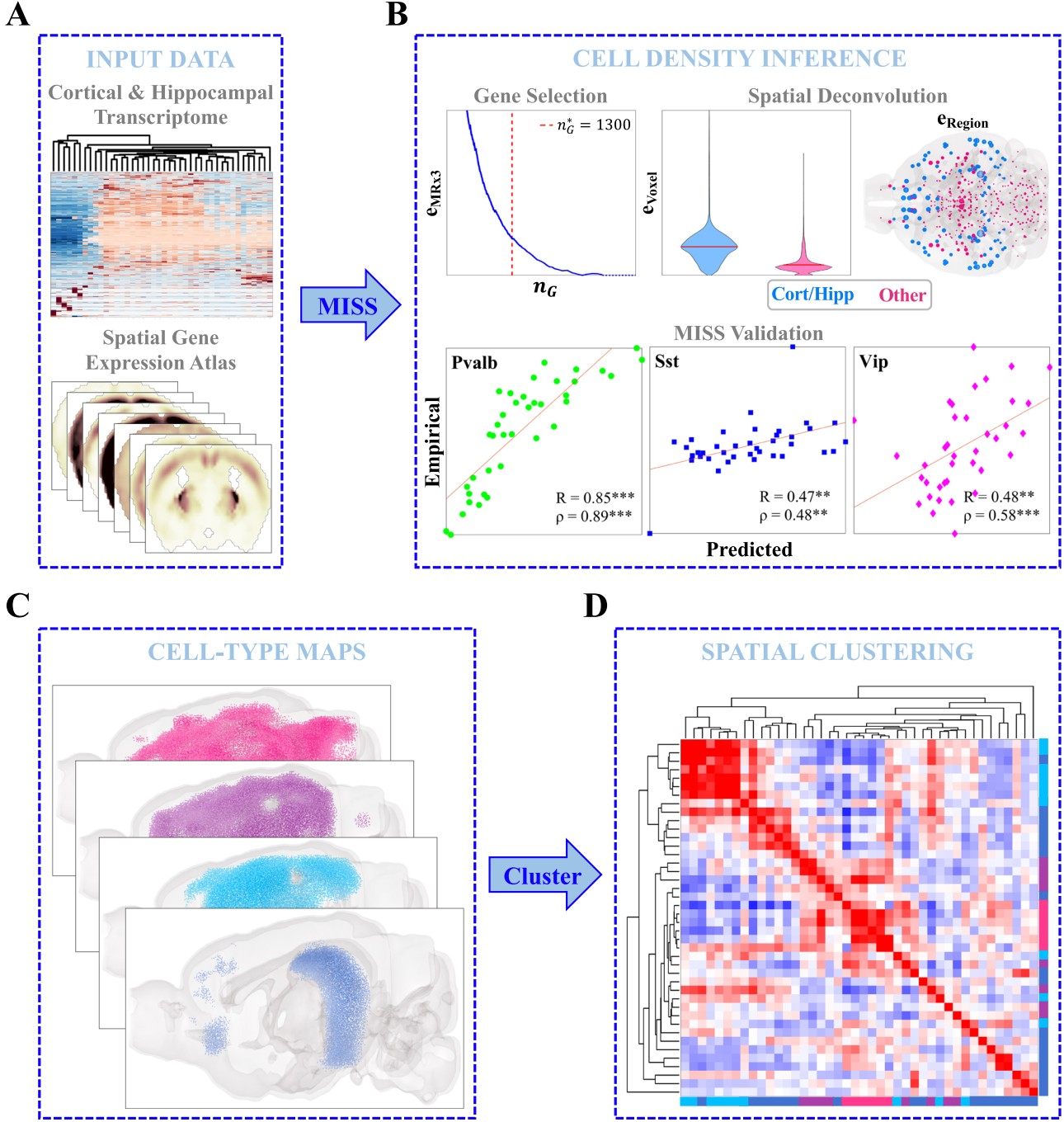

**Fig. 1 | Matrix inversion and subset selection (MISS). A** Neocortical and hippocampal scRNAseq data from Yao, et al.[52] and the Allen Gene Expression Atlas (AGEA)[58] were used to infer cell-type densities using the Matrix Inversion and Subset Selection (MISS) pipeline[49]. **B** MISS proceeds in two parts: informative gene selection and spatial deconvolution. The first step involves choosing an informative gene subset using the MRx3 subset selection algorithm (see Methods). Here we found an optimal 1300-gene subset using a similar procedure to that previously published (*top left*). The second step involves solving a nonnegative least-squares problem voxel-by-voxel to determine the whole-brain densities of the Yao, et al. cell types. The top center and top right panels show that the algorithm did not exhibit greater degrees of residual error inside of neocortical and hippocampal voxels and regions than elsewhere. We validated our results using published interneuron distributions in the neocortex[91], demonstrating that MISS is accurately inferring cell-type densities (*bottom panels*). **C** MISS allowed us to create whole-brain maps of cell types at a resolution of 200 μm. **D** Spatial clustering of these distributions at a regional level reflected a remarkable amount of diversity between cell types.

## Tauopathy dataset curation

We assembled a set of 12 regional tauopathy datasets, each of which represents a unique experimental condition. Here, we define an experimental condition in terms of the genetic background of the mice used, the type of injectate used, and the injection site (see Table S3 for brief descriptions of these datasets)[39–43]. We then coregistered the inferred MISS cell-type maps, which were parcellated using the Common Coordinate Framework version 2 (CCFv2)[59], into the region spaces of the individual datasets and performed a combination of univariate and multivariate analyses. This meta-analysis of mouse tauopathy allows us to answer broad questions about cell-type selective vulnerability at a whole-brain level. For visualizations of the end-time-point distributions of tau pathology for each of these datasets, refer to Fig. S7.

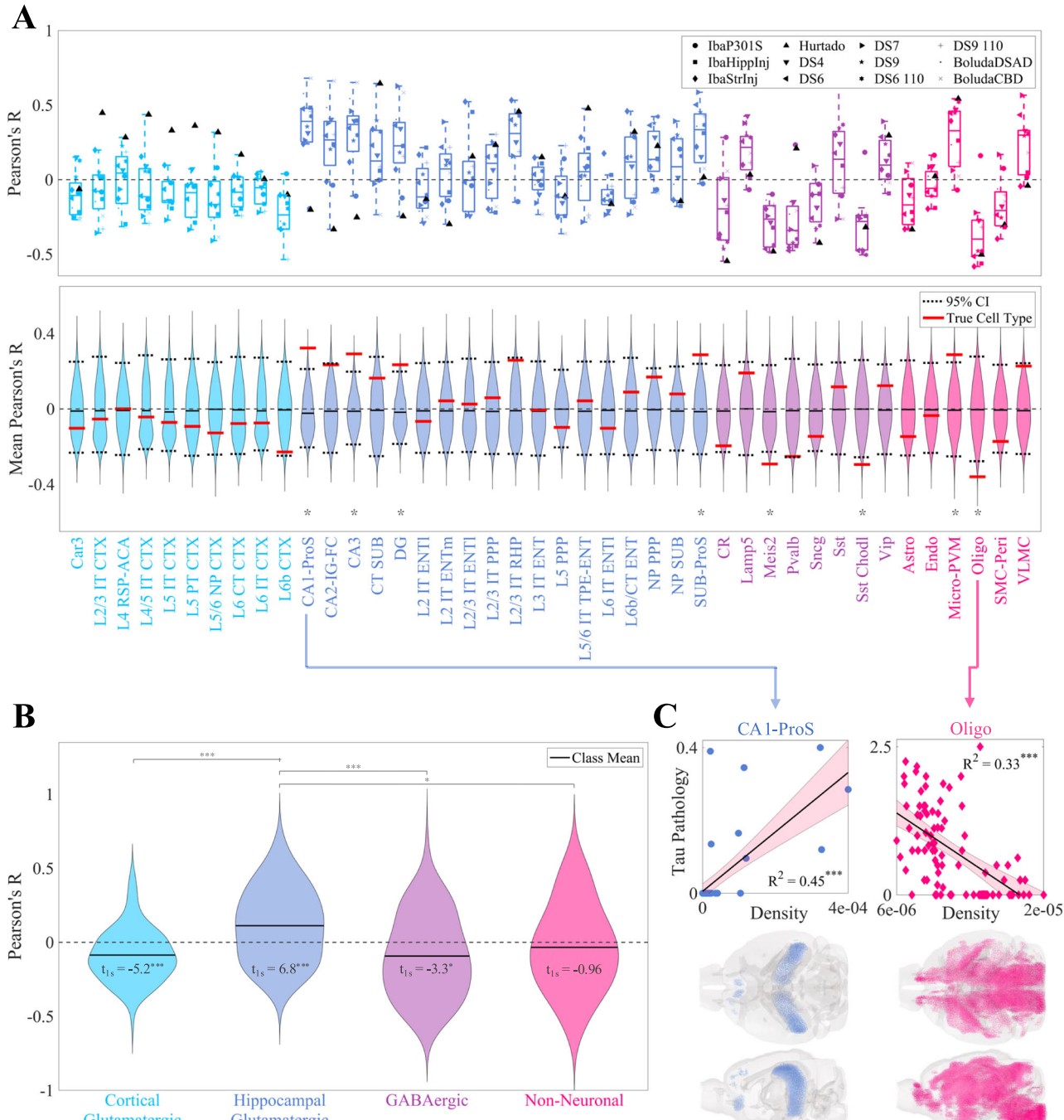

**Fig. 2 | Univariate analysis of the relationship between cell-type density and end-time-point pathology across mouse tauopathy models. A** (*Top*) Box plot of Pearson's correlation coefficients between the MISS-inferred distributions of the 42 cell types of the Yao, et al. dataset and the end tau pathology for 12 mouse tauopathy datasets. (*Bottom*) Violin plots of the mean Pearson's R values for autocorrelation-preserving spatial null models for each cell type. Each distribution represents the mean correlation across datasets for 10,000 spatial null instances. The violins are shown alongside their 95% confidence intervals (black dashed line) and the mean R values of the actual cell types distributions (red lines). *$p < 0.05$. **B** Violin plots of the data in (**A**), where cell types have been grouped into four general classes. One-sample

t-statistics are displayed within each violin plot and the statistically significant pairwise comparisons are bracketed. **C** Scatter plots of the single-strongest associations among datasets for the cell type with the highest mean positive correlation (CA1-ProS, a glutamatergic hippocampal, with the BoludaCBD dataset) and the cell type with the highest mean negative correlation (Oligo, oligodendrocytes, with the IbaStrInj dataset). Also shown are glass brain representations of the whole-brain distributions of these two cell types. *$p < 0.01$; **$p < 0.001$; ***$p < 0.0001$. Refer to the original publication for a full description of the cell-type annotations[52] and Table S3 for a summary of the tauopathy datasets.

## Identifying vulnerable and resilient cell types

Figures 2A and S8 depict Pearson's correlations (R values) and *p*-values between the spatial distributions of cell types and the patterns of end-time-point τ deposition across the 12 experimental conditions, respectively. Here, a positive R value indicates that a cell type may be more vulnerable to later

invasion of τ pathology (i.e., secondary cellular vulnerability)[12], while a negative value suggests that it may confer resilience. To more thoroughly assess the statistical significance of each cell type's association with τ pathology across datasets, we generated distributions of mean R values using spin null models of cell-type densities generated using the BrainSMASH

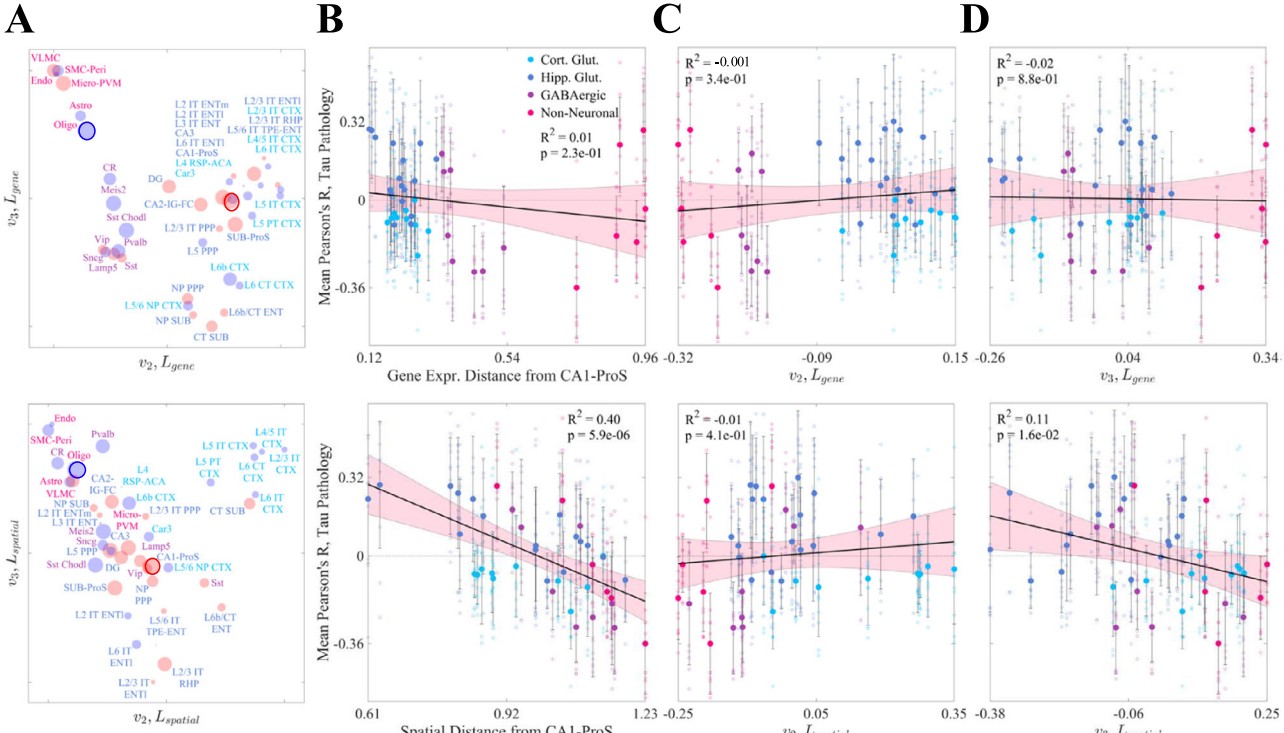

**Fig. 3 | Spectral embedding analysis. A** All cell types in the Yao, et al. dataset plotted in terms of their projections on $v_2$ and $v_3$, the second and third eigenvectors of the Laplacian of the gene expression correlation similarity matrix ($L_{gene}$, top panel) and of the Laplacian of the spatial correlation similarity matrix ($L_{spatial}$, bottom panel). The radius of each point corresponds to the magnitude of that cell type's mean correlation to $\tau$ pathology and the color indicates the sign (red = positive, blue = negative). **B** Linear models showing the relationships between end-time-point tau pathology and gene expression or spatial distance from CA1-ProS for all other cell types in the

Yao, et al. dataset; only spatial distance is significantly associated with tau. The pink shaded area represents the 95% confidence interval for the least squares regression. **C, D.** Linear models showing the relationships between $v_2$ (**C**) and $v_3$ (**D**) of $L_{gene}$ (top panels) and $L_{spatial}$ (bottom panels) with end-time-point $\tau$ pathology for all cell types in the Yao, et al. dataset. Only $v_3$ for the spatial Laplacian shows a significant relationship with pathology at a $p < 0.05$ threshold. Refer to the original publication for a full description of the cell-type annotations[52]. Error bars correspond to ± 1 standard deviation above/below the mean.

toolbox[60] (Fig. 2A, bottom panel; see also Methods). 8 of the 42 types exhibited significant associations ($p < 0.05$): CA1-ProS, CA3, DG, Meis2, Micro-PVM, Oligo, SUB-ProS, and Sst-Chodl (asterisks).

At the level of cell-type classes, $\tau$ pathology showed a positive correlation with hippocampal glutamatergic neurons ($p = 4.3 \times 10^{-10}$, Bonferroni-corrected) and negative correlations with both cortical glutamatergic and GABAergic neurons ($p = 3.3 \times 10^{-6}$ and $p = 4.3 \times 10^{-3}$, Bonferroni-corrected, respectively) (Fig. 2B). Among these, the hippocampal glutamatergic cell type CA1-ProS stood out as the single-most vulnerable cell type (Mean R = 0.32), which is entirely confined to the hippocampal formation and in particular CA1 (Fig. 2C). While non-neuronal cell types generally lacked a significant correlation to $\tau$ pathology (Fig. 2B; Table S4). Oligo (oligodendrocytes) were the most resilient cell type, with a mean R of −0.36 (Fig. 2C), and the immune cell subtype comprising microglia and perivascular macrophages (Micro-PVM) exhibited a significant positive correlation (mean R = 0.29).

Similar results and significant cell types were obtained when assessing these associations with the non-parametric Spearman's $\rho$ instead of Pearson's R (Fig. S9). Repeating this analysis on earlier time points also yielded results that resembled those of the final one (Fig. S10); this aligns with the observation that there is some overlap between $\tau$ pathology distributions at different time points within the same study (Fig. S11). The significance of these correlations, especially considering the known roles of these glial cells in disease progression, is further elaborated in the Discussion.

**Spectral embedding analysis reveals that SV-C and SR-C are not determined by gene expression**

To further understand the structure of these cell-type data, we created two similarity matrices: $S_{gene}$ for gene expression and $S_{spatial}$ for spatial

distribution. Each entry $(i, j)$ of each matrix denotes the transcriptomic or spatial correlation-based similarity between cell types $i$ and $j$, respectively. We then performed spectral embedding, which involves the eigencomposition of the Laplacian matrices derived from $S_{gene}$ and $S_{spatial}$, and projected each cell type onto the first two nontrivial eigenvectors, $v_2$ and $v_3$ (Fig. 3A; see also Methods). These eigenvectors represent a low-dimensional projection of cell types in terms of their gene expression and spatial distributions, respectively.

In gene expression space, the cell types generally fell into three clusters: non-neuronal cells, GABAergic neurons, and glutmatergic neurons (Fig. 3A, *top*; Fig. S2A). The evident overlap between hippocampal and neocortical glutamatergic neurons suggests that the delineation between these cell types does not strictly follow regional boundaries, aligning with the findings from Yao, et al.[52]. Clustering was much less distinct when looking at spatial similarity, with one small group of primarily neocortical glutamatergic neurons separating from the rest of the cell types (Fig. 3A, *bottom*; Fig. S2B).

Delving more deeply into the question of how gene expression and spatial similarity between cell types may relate to distributions of $\tau$ pathology, we gauged each cell type's transcriptomic and spatial divergence from the most vulnerable subtype (CA1-ProS) using a correlation dissimilarity measure, defined as $1 - S_{gene}$ or $1 - S_{spatial}$, respectively. We hypothesized that spatial distance from CA1-ProS would have a stronger relationship to $\tau$ distributions than gene expression distance; indeed, when we attempted to use these distances to fit $\tau$, we found a significant negative association for spatial distance and none for gene expression distance (Fig. 3B). These results persisted at a more global level as well: there was no association for $v_2$ or $v_3$ of $L_{gene}$ with $\tau$ pathology (Fig. 3C and D, *top*), but a statistically significant, albeit weaker association for $v_3$ of $L_{spatial}$ (Fig. 3C and D, *bottom*).

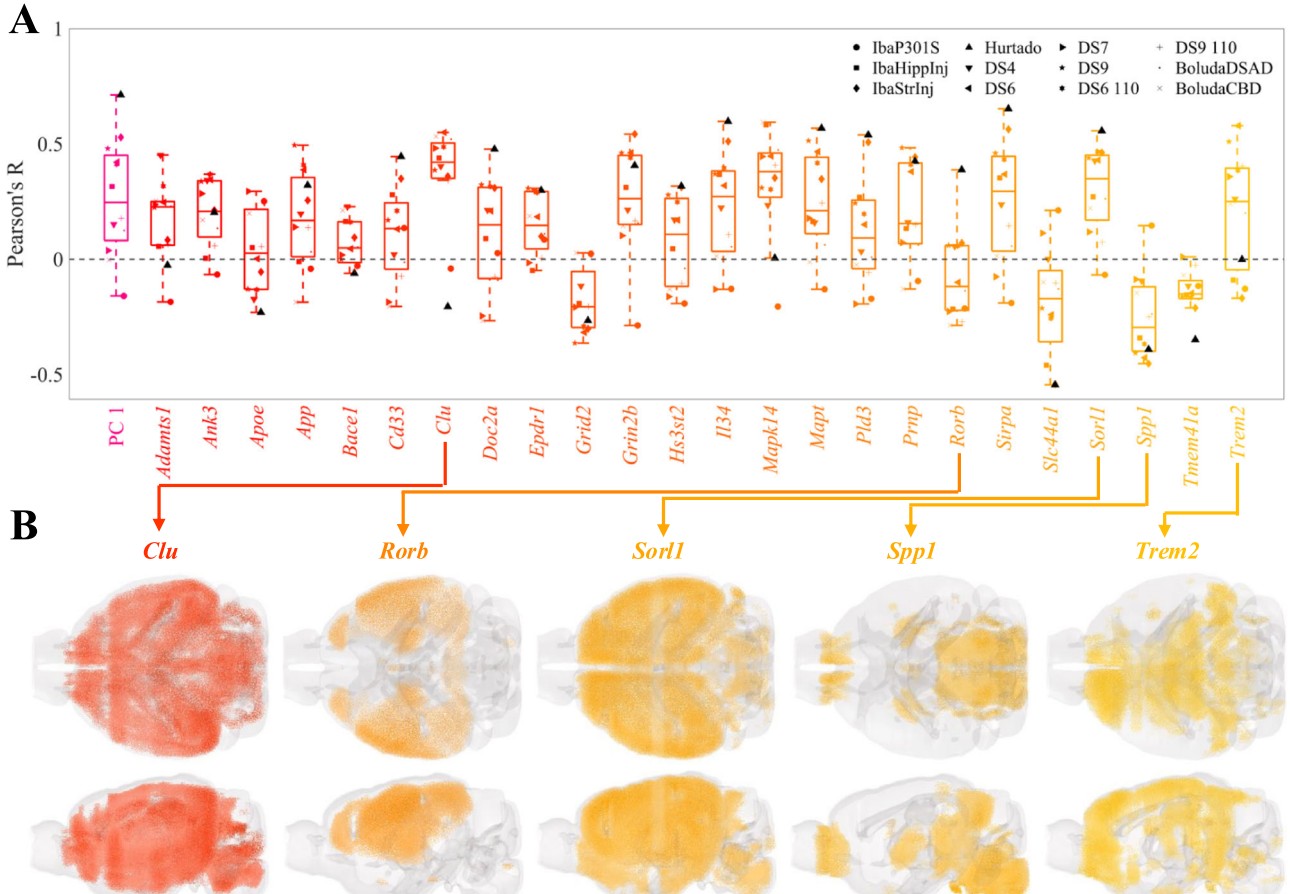

**Fig. 4 | Univariate analysis of the relationship between AD risk genes and end-time-point pathology across mouse tauopathy models. A** Box plot of Pearson's correlation coefficients between the MISS-inferred distributions of the 24 AD risk genes and the end tau pathology for 12 mouse tauopathy datasets. The first principal component (PC 1) of these genes is shown to the left in magenta. The list of AD risk genes is provided in Table S5. **B** Glass brain renderings of genes of interest.

These results suggest that vulnerable cells are not clearly distinguished from non-vulnerable cells by their molecular composition alone.

## Correlating regional distributions of AD risk genes with tau pathology

We next examined the regional vulnerability or resilience to $\tau$ in terms of the expression of 24 known AD risk, which we curated by literature search (Table S5)[53,54]. Fig. 4A shows the variability in correlations between regional gene expression and $\tau$ pathology across tauopathy datasets and individual genes. The clusterin gene (*Clu*) stands out as the most correlated with $\tau$ (mean R = 0.36) and exhibits broad expression throughout the mouse brain, which is especially strong in the hippocampal regions (Fig. 4B). By contrast, the most anti-correlated gene, osteopontin (*Spp1*; mean R = −0.25) is primarily expressed in the hindbrain and olfactory areas, regions largely unaffected by $\tau$ pathology. Other genes also warrant special attention for their previously noted contributions to selective vulnerability. The nuclear receptor ROR-beta (*Rorb*) gene, a marker for layer-4 neocortical neurons that also distinguishes a subset of especially vulnerable EC neurons[11], is broadly expressed throughout layer 4 of the neocortex (Fig. 4B). However, with the notable exception of the unseeded Hurtado study[40], correlations to $\tau$ pathology were low in magnitude and trended negative, similar to the neocortical glutamatergic neurons (Fig. 2A). The neuronal sorting receptor gene *Sorl1*, by contrast, exhibited notable and consistent correlations to $\tau$ pathology and is broadly expressed throughout the forebrain (Fig. 4B). *Trem2*, a marker for disease-associated microglia (DAM), also generally aligned with overall $\tau$ distributions. We also found the association between $\tau$ pathology and the AD-risk eigengene, which we

defined as the first principal component of these 24 genes; it exhibits a variable, net-positive association across datasets (R = 0.27).

## Cell types better predict regional vulnerability than AD risk genes

We next regressed brain-wide $\tau$ pathology distributions using cell-type maps as linear predictors. We used a Bayesian Information Criterion (BIC)-based approach for model selection to control for overfitting, ensuring that each linear model was constructed with only those cell types that were maximally informative (Fig. S12; see also Methods). We found that all models exhibited statistical significance of at least $p < 0.0001$ (Fig. 5A; Table 1). Additionally, we found a remarkable diversity among the informative cell types (Fig. 5B). Oligodendrocytes were selected the most frequently at a rate of 50%, followed by the microglia/perivascular macrophage cell type and three hippocampal glutamatergic neurons (CA1-ProS, CA3, and SUB-ProS). More generally, the four cell-type classes were not equally represented among the informative cell types ($\chi^2_{GOF} = 15.0$, $p = 5.5 \times 10^{-4}$). Notably, the most underrepresented by overall frequency was cortical glutamatergic neurons, with only three cell types selected once each (L2/3 IT CTX, L5/6 NP CTX, and L6b CTX). To ensure that the BIC-based model selection procedure did not bias our results, we repeated these analyses using the five most correlated cell types per dataset and obtained qualitatively and quantitatively similar results (Fig. S13; Tables S6 and S7). We also performed 10-fold cross-validation on the BIC-selected cell types per dataset to more robustly assess how well the linear models fit the data in the context of overfitting. While performance degraded as expected, all but 3 datasets retained significance at a $p < 0.01$ level (Fig. S14; Table S8).

When we attempted to model $\tau$ pathology as a function of AD risk genes using a similar variable selection procedure, we found that the cell-

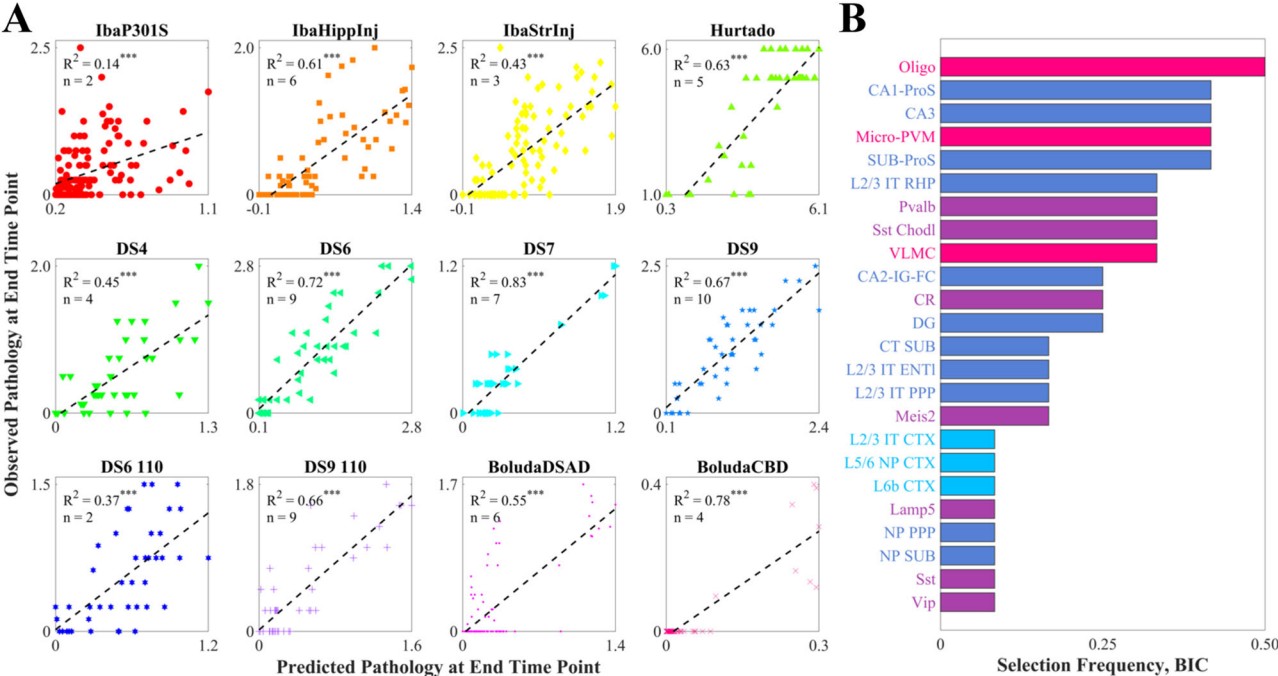

**Fig. 5 | Multivariate analysis of end-time-point pathology. A** Scatter plots of the optimal cell-type-based models of tau pathology at the end time points for each of the nine mouse tauopathy studies, along with their associated R² values and the number of cell types chosen using the BIC. *$p < 0.01$; **$p < 0.001$; ***$p < 0.0001$.

**B** Bar plot of the frequency with which cell types were included in the BIC-based linear models in (**A**). Of the 42 cell types in the Yao, et al. dataset, 24 were selected at least once.

type-based models had lower BIC values than their gene-based counterparts for 11 of the 12 tauopathy datasets, indicating that the former were generally superior (Table 1; Figs. S15 and S16). While all AD gene models reached a $p < 0.01$ significance level, several – most prominently, DS7 and DS9 110— were more weakly significant than their cell-type counterparts. We also found that cell-type-based linear models universally outperformed AD-risk-gene models when using the five best correlates as predictors (Fig. S17 and Table S6), demonstrating that these results are robust to variable selection method. When we repeated the 10-fold analysis described above (Fig. S14), AD genes still generally performed worse (though less markedly so) than cell types. The AD-gene models had higher (worse) BIC values for 7 out of the 11 datasets for which at least one of the two models reached the

$p < 0.01$ significance threshold, although they reached this threshold for one more dataset (10 of 12) than the cell-type models (9 of 12) (Fig. S18; Table S8).

### Regional and cell-type vulnerability may change over the progression of the disease

The above analyses, given that they rely on end-time-point $\tau$ pathology, attempt to answer questions about *secondary* cellular vulnerability and resilience; that is, they explore on cell types that may confer regional susceptibility to $\tau$ invasion once pathology has already started[12]. However, there is also significant interest in identifying cell types that play a role in *primary* cellular vulnerability, which relates to the initiation of the pathological process. Among the 12 tauopathy datasets, the Hurtado et al. mouse model[40] is unique, because it exhibited $\tau$ pathology entirely endogenously and therefore does not the have the confound of an injected seed. This mouse model therefore provided an opportunity to study early tangle accumulation without the influence of external seed injections. Regressions on pathology data from different time points yielded significant associations with our model predictions (Fig. 6A; Fig. S20; Table S9). Both observed and predicted $\tau$-pathology also agree visually, capturing early regional involvement and its subsequent spread (Fig. 6B, C). Interestingly, the dominant feature in the linear models shifted across time points. At 2 months, where only caudal neocortical and entorhinal regions exhibited mild $\tau$ pathology, the EC-localized L3 IT ENT neuronal subtype was the most informative (Fig. 6D, left panel). By contrast, Sst and CT SUB neurons, both mainly found in the forebrain, were the most informative at 4 and 6 months, respectively (Fig. 6D, middle panels). At the final time point, oligodendrocytes stood out as the most significant cell type (Fig. 6D, right panel); we note that the oligodendrocyte distribution is anti-correlated with $\tau$ for this mouse model (Fig. 2A).

Although a true assessment of primary cellular vulnerability cannot be assessed in seeded mouse models (see Discussion), we sought to systematically explore early stage vulnerability by repeating the analyses shown in Fig. 5 for the first and second time points in each dataset, in analogous fashion to the univariate analyses presented in Fig. S10. Although all linear

### Table 1 | BIC linear regression model statistics

| Dataset | Cell types (BIC) | | | AD risk genes (BIC) | | |
|---|---|---|---|---|---|---|
| | R² | n | BIC | R² | n | BIC |
| IbaP301S[42] | 0.14*** | 2 | 180.2 | **0.28*** | **8** | **176.4** |
| IbaHippInj[41] | **0.61*** | **6** | **81.0** | 0.43*** | 3 | 109.0 |
| IbaStrInj[41] | **0.43*** | **3** | **163.5** | 0.31*** | 1 | 174.4 |
| Hurtado[40] | **0.63*** | **5** | **153.3** | 0.41*** | 1 | 163.8 |
| DS4[43] | **0.45*** | **4** | **50.6** | 0.35*** | 2 | 52.5 |
| DS6[43] | **0.72*** | **9** | **79.5** | 0.53*** | 2 | 84.5 |
| DS7[43] | **0.83*** | **7** | **−26.9** | 0.21*** | 1 | 23.2 |
| DS9[43] | **0.67*** | **10** | **70.3** | 0.38*** | 2 | 75.4 |
| DS6 110[43] | **0.37*** | **2** | **47.2** | 0.32*** | 2 | 50.4 |
| DS9 110[43] | **0.66*** | **9** | **49.9** | 0.22*** | 2 | 67.1 |
| BoludaDSAD[39] | **0.55*** | **6** | **68.0** | 0.34*** | 2 | 87.8 |
| BoludaCBD[39] | **0.78*** | **4** | **−167.7** | 0.54*** | 4 | −127.3 |

Statistics corresponding to the linear models shown in Fig. 5 and Fig. S15. Bold font indicates the best model by the Bayesian Information Criterion (BIC). *$p < 0.01$; **$p < 0.001$; ***$p < 0.0001$.

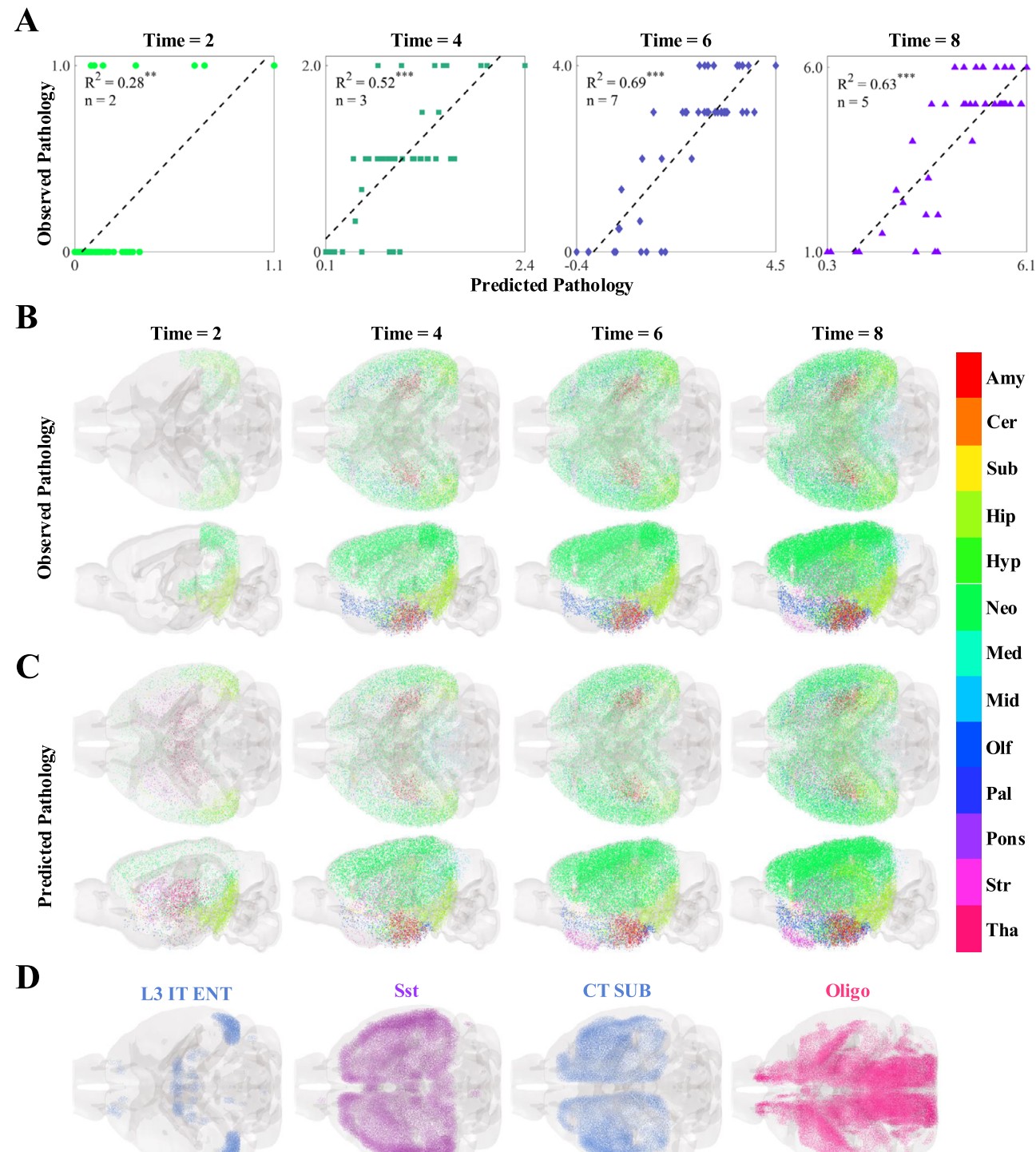

**Fig. 6 | Multivariate analysis of the temporal progression of pathology in an unseeded mouse tauopathy model. A** Scatter plots of the optimal cell-type-based models of tau pathology for each time point in the unseeded Hurtado, et al. mouse tauopathy dataset[40], along with their associated $R^2$ values and the number of cell types chosen using the BIC. Time is given in months. *$p < 0.01$; **$p < 0.001$; ***$p < 0.0001$. **B, C** Glass-brain representations of the observed **B** and predicted **C** pathology plotted along the y-axis in (**A**) over time. The color indicates major region-group: Amy amygdala, Cer cerebellum, Sub cortical subplate, Hip hippocampus, Hyp hypothalamus, Neo neocortex, Med medulla, Mid midbrain, Olf olfactory, Pal pallidum, Pons pons, Str striatum, Tha thalamus. **D** Glass-brain representations of the voxel-wise distributions of the four cell types that appear in the BIC-optimal linear models in (**A**) for all four time points. L3 IT ENT layer-3 intratelencephalic entorhinal neuron, Sst telencephalic somatostatin-expressing GABAergic neuron, CT SUB Corticothalamic neuron of the subiculum, Oligo oligodendrocytes. Please refer to the original publication for a full description of the cell-type annotations[52].

models at earlier time points exhibited statistical significance at the $p < 0.001$ level (Fig. S20A and C), the cell types that were selected were notably different (Fig. S20B and D). In particular, we found that, at time point 1, the two cell types with the strongest overall correlations to $\tau$ (Fig. 2A) and that

were among the most highly represented in the end-time-point linear models (Fig. 5B), CA1-ProS and Oligo, were no longer selected nearly as frequently (Fig. S20B). CA1-ProS only appeared in linear models for 1 out of the 12 datasets, while Oligo was not selected at all. While we underscore that

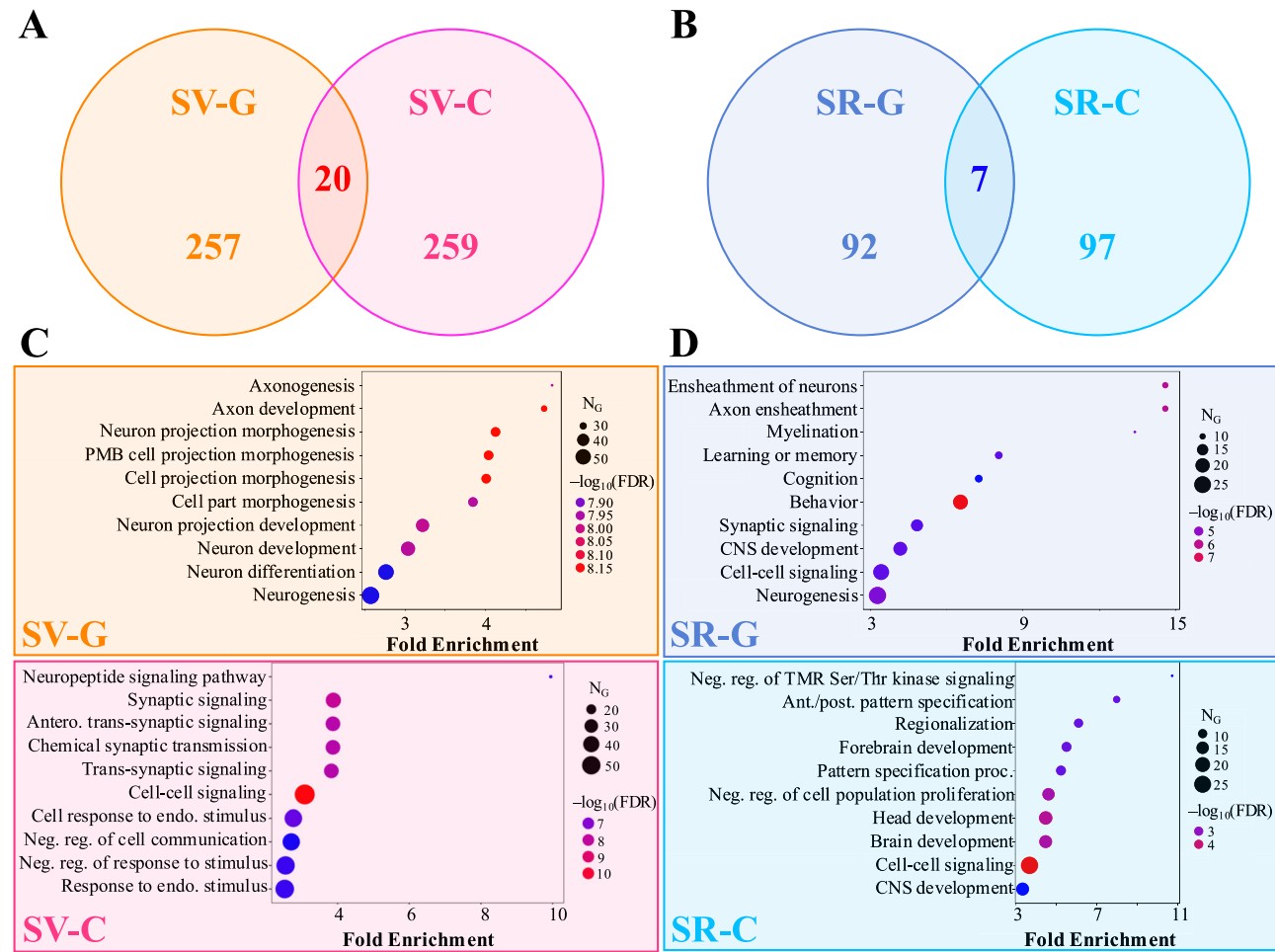

**Fig. 7 | Gene ontology analysis of vulnerable and resilient gene sets. A** Venn diagram of the gene-based selective vulnerability (SV-G) and cell-type-based selective vulnerability (SV-C) genes. **B** Venn diagram of the gene-based selective resilience (SR-G) and cell-type-based selective resilience (SR-C) genes. **C** Top 10 biological processes by fold enrichment represented in the SV-G (*top*) and SV-C (*bottom*) gene sets. **D** Top 10 biological processes by fold enrichment represented in the SR-G (*top*) and SR-C (*bottom*) gene sets. PMB plasma membrane bounded, TMR transmembrane receptor, CNS central nervous system.

studying primary cellular vulnerability in these mouse models is confounded by seeding effects, these results align with the temporal shift for Oligo observed with the unseeded dataset and suggest the possibility that oligodendrocytes may participate at later stages of disease.

**Distinct mechanisms underlie gene and cell-type-based selective vulnerability and resilience**

To more deeply explore similarities and differences between cell-type-based and gene-based selective vulnerability and resilience, we identified four distinct sets of genes: (1) *SV-G*: Top 10% of genes positively correlated with $\tau$ across datasets; (2) *SV-C*: Top genes differentially expressed in vulnerable cell types; (3) *SR-G*: Top 10% of genes negatively correlated with $\tau$ across datasets; and (4) *SR-C*: Top of genes differentially expressed in resilient cell types. The sizes of the SV-C and SR-C gene sets were matched to the sizes of the SV-G and SR-G gene sets, respectively, to enable comparisons between them (see Methods for more details on the gene selection procedure).

We found minimal overlap between the SV-G and SV-C gene sets or the SR-G and SR-C gene sets (Fig. 7A and B). We also note that, among the 24 AD risk genes explored above (Table S5), only *Spp1*, *Cd33*, and *Trem2* overlapped with any of these four gene sets; *Spp1* was a part of the SR-G set and *Cd33* and *Trem2* were in the SV-C set. A subsequent gene ontology (GO) analysis revealed that these gene sets were also associated with distinct biological processes, albeit with some overlap. For instance, SV-G genes were predominantly associated with neuronal development, whereas SV-C

genes were more associated with synaptic processes and cell signaling (Fig. 7C). Similarly, the SR-G gene set was enriched in genes involved with axon maintenance and cognition, while SR-C gene set was linked to the macroscopic organization of the central nervous system during development (Fig. 7D). Similarly, we found little overlap between GO-identified cellular components or molecular functions for these four gene sets (Fig. S21).

Since there was a notable discrepancy between the associations across different classes of glutamatergic neurons (Fig. 2B), we sought to probe the molecular mechanisms that may underpin these differences. Using the same measure of differential expression as with the **SV-C/SR-C** gene sets, we obtained the 100 most differentially expressed genes between cortical and hippocampal glutamatergic neurons and performed GO analysis as before (Fig. S22). The biological processes, cellular components, and molecular functions were all distinct between these two classes. The hippocampal glutamatergic genes exhibited processes associated with synaptic function and regulation of GABAergic signaling, while the cortical glutamatergic genes were a part of networks that were largely not specific to the central nervous system. While it is likely that these genes are incompletely annotated for all cell types and tissues, the fact that hippocampal glutamatergic neuron genes are involved with synaptic processes is intriguing, given that this class is positively associated with $\tau$ as a whole.

Overall, these findings underscore that the genetic and mechanistic bases of selective vulnerability and resilience differ substantially from the perspective of cell types versus genes.

## Discussion

Differential susceptibility between regions to pathological protein species like $\tau$ is a hallmark of AD and many other neurodegenerative diseases. The discovery of neuronal subtypes in regions that are disproportionately afflicted by $\tau$ pathology has given credence to the idea that regional susceptibility may depend on its cell-type composition[9,11,12,14]. Here, we took a computational, meta-analytical approach towards exploring this cell-type-based selective vulnerability (SV-C) and resilience (SR-C) across the whole brain, leveraging available regional tauopathy data in twelve PS19 mouse models[39–43]. To our knowledge, this is the broadest exploration of selective vulnerability at a whole-brain level to date. After inferring the baseline whole-brain distributions of 42 subclasses of cell types using the recently developed Matrix Inversion and Subset Selection (MISS) algorithm[49,52] (Figs. S3–S6), we performed various statistical analyses to answer the following questions: (1) Which individual cell types and cell-type classes contribute most to SV-C and SR-C? (2) How well are the distributions of $\tau$ pathology explained by baseline cell-type distributions? (3) How do SV-C and SR-C compare to the selective vulnerability and resilience conferred by AD risk genes? (4) Are SV-C and SR-C related to the same genes and functional gene networks as gene-expression-based selective vulnerability (SV-G) and resilience (SR-G)?

Among the four classes of cell types present in the Yao, et al. dataset, only hippocampal glutamatergic neurons showed a significantly positive association with $\tau$ pathology (Fig. 2B). Conversely, we found net negative associations across cortical glutamatergic neurons and GABAergic neurons and no association with non-neuronal cell types. The discrepancy between the two classes of glutamatergic neurons is especially noteworthy given the fact that these types do not cleanly separate in gene expression space (Fig. S2)[52]. Underscoring this result more broadly is the surprising finding that there was no association between $\tau$ pathology and the spectral eigenvectors of the gene expression profiles of the Yao, et al. cell types (Fig. 3). Therefore, although we were able to draw broad conclusions about which classes of cells may contribute to SV-C and SR-C, a given cell type's gene expression cannot easily explain whether that type will exhibit vulnerability or resilience to $\tau$ pathology.

To confirm the impression that observed SV-C and SR-C were not simply a direct consequence of the identified cells' gene expression, we performed several statistical comparisons. We found that SV-C and SR-C were significantly stronger effects than the SV/SR related to GWAS-identified AD risk genes (Fig. 5; Table 1). Furthermore, genes that mapped onto vulnerable cells were involved in quite distinct biological processes compared with genes directly associated with vulnerability or resilience (Fig. 7). These findings shed light on the distinct roles played by genes and cell types, and have the potential to help explain the dissociation between upstream genes and downstream pathology common to many neurodegenerative diseases[3,4].

Below we discuss several notable individual cell types in light of their contributions to SV-C or SR-C. These are broad findings that were found to generalize across studies. We emphasize that a high degree of variability exists between individual cell types within each class and between datasets (Fig. 2A, B). No single correlation between any dataset's end-time-point $\tau$ pathology and any cell type was higher than 0.7 in magnitude, nor were any mean correlations across datasets greater than 0.4. That the associations were modest-to-moderate in strength suggests that a more nuanced understanding of the cellular underpinnings of $\tau$ vulnerability would require detailed mechanistic investigation of specific cell types identified here[11,14]. We further note that an assessment of the significance of pairwise differences between cell types could only be performed at the class level Fig. 2B) because of the small number of datasets relative to the number of individual types. Nevertheless, we make mention of cell types that stood out both in the univariate (Fig. 2) and multivariate (Fig. 5) analyses.

The CA1-ProS neuron, among the 42 cell types evaluated, exhibited the strongest mean positive correlation with $\tau$ pathology across the 12 tauopathy datasets (Fig. 2A) and was one of the most frequently selected cell types in the multivariate models (Fig. 5B). We note that this association

persisted after removing the seeded regions from consideration, indicating that it is not being directly driven by seeding site. Although the present work is only correlative and therefore limits the conclusions we can draw on a mechanistic level, this finding is notable because CA1 neurons have been previously identified as being especially susceptible to energy deprivation in APP/PS1 mouse models, especially when glucose and oxygen supply is compromised[61,62]. CA1 neurons have also been identified to be particularly vulnerable to hyperphosphorylated $\tau$ in mouse models using in situ cell-type mapping[63].

Neurons isolated from entorhinal regions, however, largely did not positively associate with $\tau$ across the 12 datasets (Fig. 2A) nor did they feature in the multivariate models (Fig. 5B; Fig. S13B), despite the fact that the EC is one of the earliest regions to exhibit $\tau$ pathology in AD[1]. By contrast, scRNAseq performed on postmortem AD patients identified glutamatergic neurons in layer II of the entorhinal cortex[9], and in particular those expressing the gene *RORB*[11], to be especially susceptible to $\tau$. We also did not find a notable positive association between the baseline expression of *Rorb* in the mouse brain and $\tau$ pathology (Fig. Fig. 4); likely, this is due to the fact that *Rorb* is also a marker of L4 glutamatergic neurons in the cortex and $\tau$ pathology in these datasets is lower in cortical areas than in limbic structures (Fig. S7). However, the unseeded Hurtado et al. A$\beta$/$\tau$ mouse model revealed pronounced early EC pathology (Fig. 6B), which did exhibit strong correspondence to EC-isolated excitatory neurons such as L3 IT ENT (Figs. 2A, 6D)[40]. Overall, this may reflect the inherent limitations of the present work to explore primary cellular vulnerability, by the nature of the datasets that are currently available (see Limitations below). It may be necessary to use mouse models that develop $\tau$ endogenously, more closely mimicking human disease conditions, in order to better study primary selective vulnerability[12] to $\tau$ pathology.

Recent studies have also identified a role for GABAergic neurons in AD pathophysiology. For instance, $\tau$-dependent GABAergic synaptic dysfunction has been associated with and AD-specific pathological changes[64–66]. More recently, the first multimodal cell atlas of AD, SEA-AD, which profiled the middle temporal gyrus (MTG) of 84 human donors with AD at a single-cell level, identified subtypes of *Pvalb*+ and *Sst*+ interneurons as being prominently affected[14]. Here, we found wide variation in SV-C and SR-C with respect to GABAergic neurons, with *Sst*+ interneurons exhibiting consistently positive correlations with $\tau$ while *Pvalb*+ interneurons showed the opposite effect (Fig. 2A). However, we note that the unseeded A$\beta$/$\tau$ mouse model[40] demonstrated positive associations with both *Pvalb*+ and *Sst*+ interneurons (Fig. 2A). Because the this study was the only to use a hybrid A$\beta$/$\tau$ mouse model, it may be possible that the patterns of $\tau$ deposition are highly influenced by A$\beta$ comorbidity. Recent work in human subjects has found that indeed there are distinct blueconformational strains of $\tau$ that are specifically observed in AD and not other tauopathic diseases[67–69]. Therefore, interneuron vulnerability to $\tau$ may also in fact be conformational-strain-specific, a hypothesis that warrants further investigation.

One of our most striking results is the pronounced *resilience* oligodendrocyte-rich regions exhibited to $\tau$; it had the single-strongest correlation with $\tau$ among all cell types (Fig. 2A) and was the most frequently selected cell type in the multivariate models (Fig. 5B). These cells, which are primarily responsible for myelin production and maintenance[70], have been documented to play a role in tauopathic diseases, but whether their role is protective or harmful remains a subject of debate. Pathway analysis of gene expression changes in human AD found that oligodendrocyte-specific modules were among the most strongly disrupted[32]. AD patients exhibit detectable white matter lesions, suggesting a direct role for oligodendrocyte dysfunction in AD-related pathological changes[71]. Furthermore, although the mechanisms underpinning $\tau$ uptake in oligodendrocytes remain poorly understood[72], oligodendrocytes strongly co-localize with $\tau$ inclusions in mouse tauopathy and may facilitate $\tau$ seeding and propagation[22]. In the context of regional vulnerability, this suggests that oligodendrocytes may help to sequester $\tau$ in earlier stages of disease, they can also also serve as $\tau$ reservoirs that may be related to inter-regional spread. Here, given the

negative association between oligodendrocyte density and $\tau$ deposition at a whole-brain level, we propose that the crucial homeostatic functions carried out by oligodendrocytes may be more easily disrupted in those regions that naturally have smaller pools of oligodendrocytes at baseline.

By contrast, we found a consistently positive association between $\tau$ pathology and Micro-PVM, the immune cell subclass of the Yao, et al. dataset (Fig. 2A). Microglial activation in response to A$\beta$ and $\tau$ is a key feature of AD pathophysiology. For instance, rare genetic variants of the microglial activation gene, *TREM2*, are associated with significantly increased risk of AD[73]. However, as with oligodendrocytes, the roles of microglia in the context of AD are complex and incompletely understood. Early in the disease process, microglia effectively clear A$\beta$ pathology in mouse models[74,75], but over time, their capacity to remove plaques is attenuated[76]. Additionally, the neuroinflammatory response mounted by microglia in response to A$\beta$, which involves the release of inflammatory cytokines and the production of reactive oxygen species, can exacerbate protein pathology and induce neurodegeneration. Suppression of microglia in 5xFAD mice prevented hippocampal neuronal loss[77], and activated microglia with internalized $\tau$ were discovered in a postmortem examination of the brains of AD patients[78]. In the context of our results, regions with high baseline levels of microglia appear to have a greater vulnerability to $\tau$ pathology at later time points, suggesting that these cells may play a mediatory rather than protective role at more advanced stages of disease. Further bench work is needed to quantify microglial levels in situ to gain a more nuanced understanding of how these cells are related to pathology at a whole-brain level.

In addition to the role played by specific cell types in the CNS, non-cell-specific molecular factors may also contribute to SV. Similar to the Yao, et al. cell types, the baseline gene expression profiles of 24 AD risk genes[53,54] were variably associated with $\tau$ pathology (Fig. 4). However, we also found that cell-type distributions consistently explained end-time-point $\tau$ pathology better than this set of risk genes (Fig. 5; Table 1). A broader investigation of the genetic underpinnings of SV-C and SR-C revealed a surprising lack of correspondence between the gene expression signatures of cell types and $\tau$ pathology (Fig. 3A–C). Furthermore, the genes most directly contributing to SV-G and SR-G (that is, those directly and most strongly correlated with $\tau$) were strikingly different from those differentiating vulnerable and resilient cell types (Fig. 7). Gene ontology (GO) analysis revealed that these gene sets were enriched in distinct biological processes: SV-G genes were predominantly associated with neuronal development, whereas SV-C genes were more associated with synaptic processes and cell signaling. SR-G genes were involved with axon maintenance and cognition, while SR-C genes with the macroscopic organization of the CNS during development.

This suggests that SV-C/SR-C and SV-G/SR-G may be independent, and may involve different processes. For instance, electrophysiological or morphological features of vulnerable neurons that are unrelated to baseline gene expression in adult mice may contribute to their propensity to accumulate $\tau$[11,21]. Another extrinsic factor that contributes to the pathophysiology of tauopathies, which we did not explore here, is the trans-neuronal spread of $\tau$ along white matter tracts. Seminal work by Clavaguera, et al. demonstrated that the injection of pathological $\tau$ was sufficient to induce misfolding and aggregation of endogenous $\tau$ in distal regions[79], and in vitro experiments have directly shown that pathological $\tau$ can travel between neurons sharing a synapse[80–84]. In this context it is especially intriguing that genes responding to vulnerable cells are functionally enriched in synaptic and cell-cell processes that may be presumed to influence trans-neuronal spread of $\tau$.

Our study carries with it several important limitations. First and foremost, while they have been invaluable in furthering our understanding of tauopathies, mouse models can only recapitulate certain aspects of human disease. For one, as discussed above, the use of exogenous seeds in most of these models inhibited us from more comprehensively studying primary cellular vulnerability and resilience at a whole-brain level. Secondly, to help minimize the numbers of different features between datasets, we relied on studies that used mice with the PS19 background (Table S3).

However, while this is a widely used model of tauopathy, the P301S $\tau$ mutation that it carries is not characteristic of AD, even if the injectates used may be derived from AD subjects (e.g., DS4 from Kaufman, et al.[43]). Further, our decision to pool these 12 datasets together allowed us to meta-analytically evaluate which cell-type features may underpin tauopathy more generally, but also introduced significant variability into our results that may have obscured more nuanced relationships. We also note that gene expression in mice differs from that in humans in important ways, some of which bear relevance to AD[85]; in part this may relate to our results concerning the *Rorb* gene (Fig. 4). However, we note that there is notable homology between cortical cell types in mice and humans despite divergent evolutionary histories[86], and gene-expression-related changes in AD have also been observed in mouse models[87].

A further set of limitations stems from the fact that we were restricted to comparing *baseline* cell-type densities – that is, those in the healthy mouse brain – with $\tau$ distributions in the disease condition. While we expect that cell-type densities are more stable and robust to disease-associated perturbations relative to gene expression, some cell types, such as microglia, have dynamic populations. Therefore, our claims can only be interpreted through the lens of *innate* regional vulnerability to $\tau$. As emerging technologies facilitate the study of disease-associated changes in gene expression, cell-type distributions, and $\tau$ deposition at the whole-brain level in the same mice (or humans presenting with tauopathy), we can more deeply delve into questions of *dynamic* regional vulnerability. Future work also includes the development of joint mathematical models of network-based spread and SV-C/SR-C to study how cell types may influence $\tau$ accumulation and spread in the context of trans-neuronal transmission. As noted above, we did not find any single cell type (or gene) that strongly explained the distribution of $\tau$ in any one study. Recently, our group developed the Nexopathy in silico (NexIS) model to examine this question in the context of gene expression[24] and this approach is easily extensible to using cell types.

## Conclusions

In summary, our study complements prior experimental approaches, offering a comprehensive exploration of the underpinnings of cell-type-specific regional vulnerability in in vivo tauopathy models. By deriving regional cell-type distributions using spatial deconvolution and then performing a meta-analysis of PS19 mouse $\tau$ pathology in the context of SV-C, we identified key cell types mediating vulnerability and resilience. We also demonstrated that SV-C is a robust mechanism for explaining diverse patterns of $\tau$ pathology across different mouse models. These results illustrate that integrative, computational approaches can reveal important insights into tauopathic disease. Further experimental work is required to elucidate specific mechanisms of vulnerability and resilience of the cell types identified here and to reconcile cell-type- and gene-associated effects with trans-neuronal spread. These efforts may prove important in the ongoing development of novel therapeutic targets.

## Methods
### Datasets

**Gene expression.** The scRNAseq data used to generate the cell-type maps come from Yao, et al. for the Allen Institute for Brain Science (AIBS), which sequenced approximately 1.3 million individual cells sampled comprehensively throughout the neocortex and hippocampal formation at 10x sequencing depth[52]. Using a standard Jaccard-Louvain clustering algorithm, the authors jointly and hierarchically clustered these samples at three taxonomic levels: class ($n = 4$), subclass ($n = 42$), and cluster ($n = 387$). The full annotation and gene expression profile of each sample, as well as trimmed mean expression across cell-type clusters, are publicly available (https://portal.brain-map.org/atlases-and-data/rnaseq/mouse-whole-cortex-and-hippocampus-10x).

Here we used this trimmed means by cluster dataset, as the Matrix Inversion and Subset Selection (MISS) algorithm only requires the consensus profiles of cell types per cluster. Utilizing the hierarchical taxonomy provided by the authors as described above, we grouped the 387 individual

clusters into subclasses as we have done previously[49], resulting in 42 unique neuronal and non-neuronal cell types spanning four major classes: cortical glutamatergic, hippocampal glutamatergic, GABAergic, and non-neuronal (Tables S1 and S2).

The spatial gene expression data come from the coronal series of the in situ hybridization (ISH)-based Allen Gene Expression Atlas (AGEA)[58]. While the sagittal atlas has better gene coverage, we chose to use the coronal atlas because of its superior spatial coverage, which provides an isotropic resolution of 200 $\mu m$ per voxel. Furthermore, MISS uses a feature selection algorithm to remove uninformative and noisy genes, partly mitigating the effect of the reduced gene coverage. We performed unweighted averaging on genes for which multiple probes were available, resulting in a dataset of 4083 unique genes. Lastly, we removed the 320 genes that were not present in both the scRNAseq and ISH datasets, resulting in a final set of 3763 genes.

**Tauopathy experiments.** We queried five studies to obtain twelve individual mouse tauopathy datasets (which we refer to interchangeably as "experiments"): BoludaCBD[39], BoludaDSAD[39], DS4[43], DS6[43], DS6 110[43], DS7[43], DS9[43], DS9 110[43], Hurtado[40], IbaHippInj[41], IbaStrInj[41], and IbaP301S[42]. We summarize the key elements of each experiment in Table S3. We selected these studies for their spatial coverage (>40 regions quantified across both hemispheres) and the fact that they all utilized the same mouse tauopathy model (PS19), which contains a P301S $\tau$ transgene on a C57BL/6 background. The only exception is the Hurtado experiment, which contained an additional mutation in the amyloid precursor protein (APP) gene. This model is particularly insightful because the endogenous $A\beta$ production induces $\tau$ pathology without the requirement of an injected seed.

**Alzheimer's disease risk gene selection.** We selected our 24 AD risk genes by finding the intersection set between the list given by the Alzheimer's Disease Sequencing Project (ADSP)[53,54] and the AGEA[58]. Gene annotations were obtained from the UniProt database[88] unless otherwise noted.

**Matrix inversion and subset selection (MISS)**
We applied the MISS algorithm to the Yao, et al. scRNAseq dataset[52] and the AGEA ISH dataset[58] as was described previously[49]. Briefly, MISS involves two steps: (1) subset selection, which utilizes a feature selection algorithm to remove low-information genes that add noise to the final prediction of cell-type density; and (2) matrix inversion, where the gene-subset spatial ISH-based gene expression matrix is regressed on the gene-subset scRNAseq-based gene expression matrix voxel-by-voxel to obtain cell-type densities. We outline each step below.

**MRx3-based subset selection.** As described previously[49], we first employed the Minimum-Redundancy-Maximum-Relevance-Minimum-Residual (MRx3) feature selection algorithm, which builds on the popular Minimum-Redundancy-Maximum-Relevance (mRMR) algorithm[89]. Let us first designate the $N_g \times N_v$ spatial gene expression matrix as $E$ (normalized by gene), the $N_g \times N_t$ scRNAseq matrix as $C$ (normalized by cell type), and the $N_t \times N_v$ target matrix of cell-type densities as $D$, where $N_g$, $N_v$, and $N_t$ are the numbers of total genes, voxels, and cell types, respectively. At this stage, $N_g = 3763$, $N_v = 50246$, and $N_t = 42$. We seek to find an informative gene subset, $S$, which survives the following procedure.

The first step of MRx3 is to remove genes from the full gene set, $G$, which contribute noise to the prediction of the spatial gene expression matrix. We use a rank-1 update rule[90] to estimate the incremental noise added per gene, as assessed by the mean-squared error between the given $E$ and the predicted $\hat{E}$, $\|E - C \cdot \hat{D}\|_2^2$), and then remove the upper 10% of genes in terms of noise added. This leaves the reduced gene set $G^*$ and the reduced matrices $E^*$ and $C^*$. We then apply mRMR on the genes of $G^*$ that remain as candidates to be added to the optimal gene set, $S$. mRMR is a greedy algorithm which, at every iteration, adds the gene to $S$ that maximizes

the following criterion $V_i$:

$$V_i = \frac{F_i}{Redund(i|S)} \quad (1)$$

$$F_i = \sum_{j=1}^{N_t} \frac{(C^*(i,j) - \overline{C^*(i,:)})^2}{N_t - 1} \quad (2)$$

$$Redund(i|S) = \frac{1}{|S|} \sum_{j \in S} |R(i,j)|. \quad (3)$$

where the index $i$ indicates the $i^{th}$ gene of $C^*$, $F_i$ is the F-statistic of gene $i$, $Redund(i|S)$ is the *redundancy* of gene $i$ with respect to set $S$, $C^*(i,j)$ is the column-normalized expression of gene $i$ of cell type $j$, $\overline{C^*(i,:)}$ is the mean expression of gene $i$ across all cell types, $|S|$ is the cardinality of $S$, and $|R(i,j)|$ is the absolute value of the Pearson's correlation between the expression of target gene $i$ and included gene $j$ of $S$ across all cell types.

This step of the MRx3 algorithm can be summarized as follows:

**Algorithm\*** **1**. **Result**: $S$, the set of $n$ MRx3-selected genes
 Initialize $S_0 = \emptyset$ and $k = 1$;
 **While** $|S| \leq n$ **do**
 $g_k = \underset{i \in G - S_{k-1}}{\mathrm{argmax}} \frac{F_i}{Redund(i|S_{k-1})}$;
 $S_k = \{S_{k-1} \cup g_k\}$;
 $k = k + 1$;
 **end**

After finding all gene sets $S_n$ of cardinality $|S| = n$ using the above procedure, we then choose the optimal value of $n$, $n_G$, that balances minimizing the reconstruction residual error and number of included genes. For the Yao, et al. dataset used in the present study, $n_G = 1300$, using the same procedure previously described[49]. This produces matrices $E_{red}$ and $C_{red}$, which only contain the 1300 rows corresponding to the genes in $S_{n_G}$. All hyperparameters of the method were consistent except that we considered sizes of $S$ from 400 to 3100 genes: $|S| < 400$ produced maps with high residuals that excluded them from consideration, and singularities in the gene expression matrix produced null predictions for $|S| > 3100$.

**Matrix inversion.** We find the densities of the 42 cell types by solving the following equation voxel-by-voxel:

$$e_{red}^k = C_{red} d^k \quad (4)$$

where $e_{red}^k$ is the $1300 \times 1$ gene expression vector for the $k^{th}$ voxel in $E_{red}$ and $d^k$ is the $42 \times 1$ cell-type density vector for the $k^{th}$ voxel in $D$. We used the `lsqnonneg` function in MATLAB to solve Equation (4).

**MISS validation.** To demonstrate the accuracy of our maps, we compared our inferred distributions of key interneurons to published regional densities in the neocortex[91]. In order to do a valid comparison between *Sst* + interneuron distributions, we averaged the densities of the Sst+ subtypes Sst and Sst Chodl in each region (Table S2).

**Coregistration and seed removal**
In order to compare the cell-type distributions to $\tau$ pathology, we required a common regional space for both. Given the whole-brain, voxel-level resolution of the cell-type maps, we were limited in this study by the spatial sampling of the $\tau$ pathology data (Table S3). As mentioned above, the regional cell-type densities were calculated by averaging across the voxel-wise densities for the 424 regions of the AIBS CCFv2[59]. Then, for each quantified region per mouse experiment, we matched the corresponding regions between these two atlases, which in most cases had a clear 1:1 correspondence. The very few regions in the tauopathy datasets that did not cleanly correspond

to the CCF parcellation were removed from consideration. In those cases where the CCF atlas contained multiple subregions of a larger region sampled in a tauopathy experiment, we averaged cell-type densities across subregions, weighted by subregion volume. Finally, each tauopathy dataset with the exception of Hurtado[40] and IbaP301S[42] were injected with a seed in a location that was uniquely defined within the CCF parcellation. For each of the other 10 experiments, we removed all seeded regions from both the tauopathy data and the cell-type maps prior to performing all downstream analyses. See Table S3 for more complete details on each tauopathy experiment.

### Statistics and reproducibility
All analyses were performed using MATLAB v.2023b. We assessed the significance of the univariate associations between end-time-point $\tau$ pathology and cell-type distributions (Fig. 2A) by constructing autocorrelation-preserving spatial null models using the BrainSMASH toolbox, which utilizes the method outlined by Burt, et al.[60] For each cell type, we generated 10,000 nulls and then calculated the correlations of each within quantified and unseeded regions for each dataset as above.

Unless otherwise stated (see, for instance, Section 2.7 of Results), multivariate linear models were fit to end-time-point tau pathology (Fig. S7) for each study individually using the MATLAB `fitlm` function. For the linear models using the distributions of multiple cell types or AD risk genes, we first performed feature selection to reduce the risk of overfitting.

Because feature selection is a nontrivial combinatorial problem and the best criterion for choosing an "optimal" model is not unambiguous, we used two different feature selection methods to construct our linear models:

1. *BIC*: We ordered the 42 cell-type features by Pearson's correlation to regional $\tau$ pathology and removed the lower 75% from consideration. For the 24 genes, we removed the lower 50% to make sure the sizes of these reduced sets were roughly the same. We then calculated the Bayesian Information Criterion (BIC) for linear models of dimensionality 2 to $n_{types} + 1$ (including the intercept term), where the features were added in descending order of Pearson's correlation. The "optimal" model for each tauopathy experiment was that which had the lowest BIC value.
2. *Top 5*: We took the five most correlated features per tauopathy experiment and constructed linear models using all of them. By definition, the "optimal" models using the second method were identical to those of dimensionality 6 using the first method.

As mentioned above, an intercept term was included for all models. Both procedures yielded qualitatively similar results in terms of model fits (through $R^2$), statistical significance, and BIC values. We also performed 10-fold cross-validation for the BIC-selected cell-type and AD-gene models using the MATLAB `fitrlinear` function, from which $F$-statistics and log-likelihoods could be calculated to determine $R^2$, $p$, and BIC values. All $p$-values provided were adjusted for false positives using the Bonferroni correction where indicated:

$$p_{corr} = \frac{p_{nominal}}{N_{tests}} \quad (5)$$

### Spectral embedding
In order to perform spectral embedding to probe the structure of the Yao, et al. cell type data we first constructed correlation distance matrices $D_{gene}$ and $D_{spatial}$, where $D = 1 - R$ and $R$ is the Pearson's cross-correlation matrix (see Fig. S1). $D_{gene}$ used the correlations between cell types in terms of their expression of the 1300 MRx3 genes (Fig. S2A) while $D_{spatial}$ used the correlations between cell types in terms of their MISS-derived densities across the 424 CCFv2 regions (Fig. S2B). We assessed the association between $\tau$

pathology and $D_{gene}$ and $D_{spatial}$, respectively, for the CA1-ProS cell type with respect to all other types.

We then constructed similarity matrices $S_{gene}$ and $S_{spatial}$ from $D_{gene}$ and $D_{spatial}$, respectively, in the following way:

$$s_{ij} = \begin{cases} \frac{1}{d_{ij}} & \text{if } i \neq j \\ 0 & \text{if } i = j \end{cases} \quad (6)$$

After min-max normalizing these $S$ matrices, we constructed the corresponding unnormalized graph Laplacians $L_{gene}$ and $L_{spatial}$, where $L = \Delta - S$ and $\Delta$ is the degree matrix of $S$. To project the cell types into the gene expression and spatial eigenspaces, we performed the eigendecomposition of $L_{gene}$ and $L_{spatial}$, respectively, and plotted each cell type's loadings on eigenvectors $v_2$ and $v_3$ (the first eigenvector, $v_1$, is trivial with an eigenvalue of 0 for Laplacian matrices). This method is very similar to the popular spectral clustering approach, which uses $k$-means clustering on these eigenvectors $v$ of $L$ to cluster data.

### Gene ontology analysis
In order to perform gene ontology (GO) analysis, we first created four gene subsets: SV-C, SV-G, SR-C, and SR-G. The SV-G/SR-G gene subsets were constructed by taking the top 10% correlated and anti-correlated genes among the 3763 in the AGEA, respectively; this resulted in a 277-gene SV-G and 99-gene SR-G subsets (Table S10 and S11, respectively). We required that the corresponding SV-C and SR-C subsets to contain equal numbers of differentially expressed (DE) genes per cell type as well as roughly equivalent sizes to these SV-G and SR-G subsets, respectively. This ensured that each cell type was roughly equally represented and that a meaningful comparison between subsets could be conducted. To this end, we first constructed the column-normalized $3763 \times 42$ matrix $C_T$, where each entry $c_T(i, j)$ represents the scRNAseq-based expression of gene $i$ in cell type $j$. We then calculated the z-scores of each gene across cell types (in other words, for each row of $C_T$). This yielded a straightforward measure of DE that could be compared across genes to identify those that best discriminated each cell type from the others.

The vulnerable cell types that were incorporated into the SV-C gene set were those that had positive mean correlations across tauopathy datasets (Fig. 2A) *and* were selected at least once in multivariate models (there were 16 such cell types; see also Fig. 5B). We took the union set of the top $n_{DEvuln}$ genes between these 16 cell types, $n_{DEvuln} = round(|\text{SV-G}|/n_{CTvuln}) = round(277/16) = 18$. After removing duplicate genes between vulnerable cell types, this resulted in a SV-C subset of 279 genes (Table S12). We used a similar procedure to construct the SR-C subset, resulting in a set of 104 genes (Table S13). Differentially expressed genes between cortical and hippocampal glutamatergic neurons were identified using the same measure, where DE was calculated relative to the glutamatergic neurons ($n = 28$) alone, and then the top 100 genes across the cell types within each class were collected (Tables S14 and S15).

We conducted separate gene ontology (GO) analyses on each of the six gene subsets using the ShinyGO toolbox[92] (v0.77, http://bioinformatics.sdstate.edu/go/) and the biological process GO database, which allowed us to identify the biological processes, cellular components, and molecular functions most functionally enriched in each gene subset. Dot plots were constructed using the built-in ShinyGO visualization tool, where we show only those biological processes that were among the top 10 by fold enrichment and also survived an FDR cutoff of 0.05.

### Visualization
All plots were generated using MATLAB v.2023b. For the 3-dimensional (3D) visualizations of tau pathology, cell-type densities, and gene expression, we utilized the in-house software package Brainframe, which creates representative point clouds from voxel-wise or region-level data, using similar procedures as previous work[24,49,93]. For visual clarity, cell-type images

were thresholded such that only voxels above the 60[th] percentile of values were displayed. We emphasize that the plotted points do not represent true locations of cells in 3D space; rather, they are randomly distributed within parcels with density proportional to their corresponding values.

## Reporting summary

Further information on research design is available in the Nature Portfolio Reporting Summary linked to this article.

## Data availability

The source data used for all analyses in this study are available on Dryad with the identifier https://doi.org/10.5061/dryad.h18931zwv[94]. Numerical data for all graphs in the manuscript can be found in Supplementary Data 1.

## Code availability

All new code for running the analyses and generating the plots can be downloaded from the following Zenodo repository (https://doi.org/10.5281/zenodo.14617270)[95] as well as on GitHub https://github.com/Raj-Lab-UCSF/CellTypeVulnerability. The code for running the MISS algorithm to generate cell-type maps from gene expression data can be downloaded here: https://github.com/Raj-Lab-UCSF/MISS-Pipeline; see also the original publication[49]. Generating the glass brain images of 3-dimensional density maps requires also the external package Brainframe, hosted here: https://github.com/Raj-Lab-UCSF/Brainframe.

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

## Acknowledgements
The authors would like to acknowledge Benjamin Sipes for his assistance with the spatial null analysis. This work was supported by the following NIH grants: R01NS092802, RF1AG062196, and R01AG072753.

## Author contributions
J.T. carried out analyses, figure generation, and drafting the manuscript. P.M. and C.A. assisted with the editing of the manuscript and study design. A.R. supervised this study and assisted with all aspects of manuscript preparation and editing.

## Competing interests
The authors declare no competing interests.
