## [Transparent Peer Review file · Communications Biology]

Searching for cellular underpinnings of the selective vulnerability to tauopathic insults in Alzheimer's disease

Corresponding Author: Dr Justin Torok

Version 0:

Reviewer comments:

Reviewer #1

(Remarks to the Author)

Torok and colleagues aimed to assess the regional association between cellular subtypes and susceptibility to tau pathology in the brain of PS19 mouse models. Cellular subtypes were identified based on single cell transcriptomics data in wild type mice provided by the Allen Institute of Brain Sciences. The mapping of tau pathology was done based on an already existing data set from multiple experimental studies on tau pathology in PS19 mice. 42 cell types identified previously by Yao et al. were mapped throughout the brain based on voxel-based MISS analyses and associated with tau pathology at the ROI level of the mouse experiments. The authors report positive and inverse association between tau and these various cell types broadly categorized according to glutamatergic, GABAergic, and non-neuronal cell types, and interpreted the findings in terms of vulnerability vs resilience. Furthermore, the authors test associations between the regional expression of AD risk genes and tau for comparison with that of the cell-types, and showed that cell-type specific correlations sometimes outperformed the AD risk genes patterns as a correlated of regional tau.

Overall, the manuscript is very well written and adds significantly to the current literature. The rationale is clear and given that the question which cellular substrates are associated with vulnerability and resilience with respect to tau pathology is still a hotly debated, the current study is timely and important. The stream of analyses which build upon each other is appealing, and framing the cell types along major axis of glutamatergic, GABAergic, and glial categories of cell types provides a useful framework. Yes, some shortcomings may be addressed as well.

Comments

Introduction:

- The authors claim that the spatial expression of AD-risk genes in the brain shows little resemblance to the regional patterns of tau deposition in patients with AD. Yet, a number of studies have reported such resemblances and - beyond AD risk genes - explored regional associations between gene expression and susceptibility to tau and Abeta based on transcriptomic data from the Allen Human Brain Atlas. The authors may provide a more balanced overview of the findings (e.g. from the Sepulcre lab, who studied that in depth).
- A strength is that data from different experimental methods to induce tau pathology in PS19 mice were taken into account to avoid confounding of regional differences in tau pathology as a result of technical idiosyncracies. Having said that, it is important to point out that tau pathology is not a uniform type of brain alteration, but systematic differences in the form of tau isoforms/strains exist that may result in marked differences in terms of cell types, distribution pattern, and phenotypes. For example, in one of the data sets from PS19 mice the authors included (Boluda et al. Acta Neuropathol 2015), tau pathology from AD patients or CBD patients were injected in PS19 mice, resulting in strong differences of tau pathology by cell type, tau spreading etc. Yet, both experimental conditions were pooled in the current study. Overall, pooling data across such meaningful differences may have watered down any cell-type specific regional differences in vulnerability to AD-like tau pathology.
- The authors may better explain their rationale for the methodological approach they took. As mentioned, there are certainly advantages in the current approach, but the disadvantages need at least to be discussed as a caveat.
- Could the authors please specify for the different tau data sets which measures of tau were used (i.e. AT8 antibody, PhF staining etc).?
- The authors reported the Pearson Moment correlations between gene expression and tau pathology (Figure 2). The cell-type dependent gene expression in the mouse brain may however be highly restricted to certain brain regions (Yao et al,

Nature 2023, Yao et al Cell, 2021), entailing potentially a non-normal distribution of gene expression when analyzed across regions. Could the authors report the distribution properties and confirm that Pearson-moment correlations rather than alternatives such as Spearman correlation are appropriate?

- The authors mention that the correlation between CA1-ProS expression and tau stood out as the single most vulnerable cell type which was highly confined to the CA1 region. (line 108). However, a visual inspection of the scatter plots suggest that other cell types of this hippocampal glutamatergic class showed a similar size of correlation, at least one that may not be significantly different from that of the CA1-ProS. Could the authors please support their stance by statistics?
- Did the authors correct for spatial auto-correlation?
- The result showing a change in directionality of the correlation coefficient between glutamatergic subtypes of the hippocampus vs isocortex was surprising. Possibly a GO analysis to interrogate the biological pathways distinguishing the effect of glutamatergic subtypes between both brain regions is useful. It is up to the authors to consider such an analysis.

Discussion:

- Overall the correlation coefficients are relatively small, in particular given the substantial variability in the size of correlation coefficients depending on the tau mouse model chosen. The authors may discuss that point.
- Please discuss differences between mouse and human gene expression. Several studies did report conserved spatial gene expression patterns between mice and humans, but also that gene expression of cell types involved in the immune response, i.e. a key factor in tau progression, may differ (e.g. Zhou et al. Nature Med 2020; Miller et al. PNAS 2010).
- Title: Given that tau pathology was assessed in the PS19 mice which carry the MAPT mutation which do not fully recapitulate the type of 4R/3R tau pathology of AD, the claim to have uncovered the cellular underpinnings of vulnerability to tau in Alzheimer's disease seems a bit strong and the title may be toned down.

Reviewer #2

(Remarks to the Author)

Torok et al. describe an analysis to investigate regional vulnerability to tau pathology by looking at atlas-based colocalization between tau, cell-type distribution and gene expression patterns in the mouse brain (specifically cortex and subcortex). The authors show that the distribution of certain cell-types and the expression patterns of certain genes show modest correlations with the distribution of tau in the mouse brain. While these findings are not validated against human single-cell atlas data, several of the findings conflict with those of such studies. The authors build inferential models trying to explain the distribution of tau as a combination of cell-types (or top genes) and, in doing so, conclude that cell-type distribution better explain tau accumulation patterns than gene expression patterns. However, the authors also show that cell- or gene-based spatial models of tau distribution seem to differ depending on at what time point during disease progression the models are formed.

The major strengths of this paper include a very creative curation, re-use and combination of datasets. The description of this data, and the degree to which it is shared is a great strength. Finally, the research question targeted by this paper is timely and of great interest to the AD field. While this paper features very interesting data, the interpretation of this data in the manuscript seems problematic. Several of the analyses contain important conceptual or methodological flaws, many large assumptions are made without being acknowledged, and at times it feels like a narrative is being pushed (in abstract and throughout manuscript) that isn't well supported by the data. These issues are expanded at length below:

MAJOR CONCEPTUAL POINTS

1) The authors show how the distribution of cell types across regions relates to tau. But how do we know that this says something about cellular vulnerability? None of the relationships are very strong in the paper, meaning specific cell types don't seem to very strongly mediate regional vulnerability. Just the fact that cell type varies by region indicates that any regional pattern should show cell-type relationships, even if weak (see null modeling point, #8 below). For example, tau is known to accumulate less in heavily myelinated areas, and heavily myelinated areas have more oligodendrocytes. The authors replicate these findings, but try to push a story about oligos mediating tau resistance, citing somewhat obscure mechanisms and papers. This association could just as likely (perhaps more likely) have to do with myelination or factors that drive it than it does with a direct role of oligos in fending off tauopathy. In fact, oligodendrocytes themselves accumulate tau pathology in 4R tauopathies like PSP, which seems to suggest against the idea that they have a natural resistance to tauopathy. This point is driven even further by the fact that linear combinations (that are not cross-validated, see point #9 below) are necessary to find strong associations in this paper — it is very likely that these models are simply learning regional patterns, not regional vulnerability. I understand that inference is limited from spatial associations, but we are talking about mouse models where direct experiments are possible at a single-cell level. Therefore, it is hard to understand why spatial association analyses in mice are needed (and interesting how many of the results seem to contradict scRNA human findings). If nothing else, this should be listed as a major limitation of the paper (there isn't even a limitations section as of now), and throughout the paper the authors really need to remove the various strongly worded sentences about mechanisms and assumptions of directionality or any kind of molecular interaction, which is not supported.

2) Similarly, throughout the paper, there is language suggesting that strong associations are seen. Some examples (my *emphasis* added):

Line 113-114: By contrast, the immune cell subtype comprising microglia and perivascular macrophages (Micro-PVM) exhibited a *strong positive correlation*.

Line 143: "Trem2, a marker for disease-associated microglia (DAM), also generally *aligned closely* with overall t distributions"

This is quite deceptive language, given that there are no strong associations suggestive of actual mechanistic interaction in any of the univariate analyses in the paper. This language must be changed.

3) Are the authors concerned that the mouse models used have a P301L background, a mutation which causes 4R tauopathies, which is a class of disease that does not include AD? Could this have something to do with the oligodendrocyte findings, since oligodendrocyte tau is a phenotype observed in 4R tauopathies, but not AD?

4) The comparison between cell types and individual genes in describing the distribution of tau produces one of the main findings described in the abstract. However, this analysis has several important flaws that cast doubt as to the authors' conclusions. First, the highest spatial correlations between tau and top AD-associated genes are equivalent to the associations with cell-type (median around 0.36), making the claim that one is superior hard to understand. Second, from a statistical perspective, the authors are not comparing apples to apples. Single transcriptomic measurements are notoriously unstable (both methodologically and biologically) and often don't replicate particularly well. However, transcriptomic networks tend to be more robust, replicable and predictive, and have a better SNR. Cell-types, even more organized, composed of many transcriptomic networks, and have even better SNR. Are the authors sure the failure of single genes to outperform cell-types in the Figure 5 analysis not just driven by the different SNR of these two measures? What if the authors compared cell-types to transcriptomic networks (generated e.g. through WGCNA or equivalent)? This would at least be a slightly fairer comparison.

5) The authors interestingly find that RORB distribution does not associate with tau pathology, contradicting actual single-cell work. However, the author themselves has recently published a study using a similar analysis in human data that found a positive association with RORB and tau distributions (10.1101/2024.03.04.583403). How do the authors reconcile these conflicting findings, or the ability of this approach in mice to be meaningful in humans?

6) It would help for the authors to make explicit how using the final timepoint affects interpretations. By using the final timepoint, the level of tau is likely somewhat saturated. This means the authors are likely assessing which regions simply do and do not (ever) get tau. This is a perfectly fine way to measure vulnerability/resilience, but it would differ from e.g. measuring vulnerability as tau arrival time. Certain regions may have lower "ceilings" of pathology (e.g. the entorhinal in cortex in humans, which is perhaps the most vulnerable but actually has relatively low levels of tau at saturation). Would the authors expect to see different results if they instead profiled vulnerability as arrival time over time, and what would that say as a measure of vulnerability/resilience? This is partially addressed in Section 2.7, but here the authors seem to change the definition of vulnerability/resilience, which confounds the discussion and interpretation the findings. If the authors trained the model on the final time point and used it to predict earlier timepoints, would it work? If not, what does that really say about this Method's ability to parse cell-based vulnerability/resilience? This section is, by the way, not covered sufficiently in the Methods to really understand what the authors have done.

7) In section 2.8, the authors appear to be surprised that genes underlying "vulnerable cell types" differ from AD risk genes associated with tau. However, top expressed genes in cell types likely have to do with cell identity and maintenance. If those genes also related to disease, you would very likely see important morphological or phenotypic differences in certain cell types, which is not a characteristic of AD. I don't fully understand the purpose behind this analysis, but the results are expected.

MAJOR METHODOLOGICAL CONCERNS

8) The ability to assess the findings described in this manuscript are greatly impeded by the fact that the underlying data is often not shown. First, it would be very helpful to show the regional (final timepoint) spatial maps of tau for the various mouse-model datasets. This will give a good sense of how variable these datasets are. Importantly, we do not know the distribution of these datasets and whether regular parametric comparisons are appropriate. Related to this point, it would be very helpful to see scatter plots (similar to Figure 5A) for the important associations in Figure 2 and 4. Visualizing them all is obviously not necessary, but perhaps the top associations. This will be helpful for establishing whether some of the associations are driven by just a few regions (which I suspect for at least some of these univariate associations, especially the finding of hippocampal neurons being most associated with pathology in Figure 2C). The nature of these points stems from concern over correlating maps with strong spatial features. It is important that the authors use spatially-preserving null models (10.1016/j.neuroimage.2021.118052) to establish how strong an association for each of these features would be expected by chance. In most cases, it will likely be non-zero.

9) Section 2.6 involves creating linear combinations of cell-types in order to predict the distribution of tau. The authors make a few statements about protection from overfitting, but BIC is unfortunately not sufficient to assess whether overfitting is happening. The real pursuit here is whether the presence of a cell-type can predict where tau accumulates. The best way to do this is through cross-validation — if one builds a model of cell-type vulnerability to tau, can it predict the distribution of tau in different data based on cell-type distribution? There are two ways the authors can do this. The first is a leave-one-dataset-out approach where the model is fit on all mouse datasets and tested on a left-out one (repeatedly) — how good is the predictive performance in such cases? The other way would be a kfold cross-validation leaving out (e.g.) 10% of regions at a time. If cell type is driving this phenomenon, a model trained on most regions should be able to predict whether tau is present in other regions. If the authors do not form this kind of cross-validated model, I am not convinced that their current models are not overfit, optimistically biased, and therefore perhaps not informative.

10) One concern I have is that we don't have a great sense how accurate the cell-type decomposition is, especially in terms of specificity. The authors show a nice validation in Figure 1B, but we do not see how well these predictions predict other

cell types. After all, as the authors show, the cell types have a great amount of transcriptomic overlap, and it is the transcriptomic information that the algorithm uses for decompositions. At the very least, it would be good to show a 3x3 matrix showing how predicted PVALB/SST/VIP relate to empirical PVALB/SST/VIP distribution. However, it would be ideal to use a few of the other decomposition algorithms out there and show that the results are similar. For example, why is it that some regions are not expressing astrocytes? Surely astrocytes are present in all brain regions?

CONCERNS WITH INTRO AND DISCUSSION

11) P1, Lines 12-14 — “Remarkably, however, the topography of vulnerable regions in AD appear to bear little relation to that of the factors that presumably cause it, especially expression of associated genes”. This is a curious comment. Many studies, including by the authors themselves, have shown spatial correlations between vulnerable brain regions and expression of key AD risk genes. Just as an example, several papers have shown MAPT expression to be associated with tau (10.1016/j.jalz.2017.02.011, 10.1093/brain/awy189, 10.1016/j.celrep.2024.113691), and APOE expression to be associated with tau accumulation in APOE e4 carriers (10.1212/WNL.0000000000011270, 10.1001/jamaneurol.2019.4421, 10.1126/scitranslmed.abl7646). Even the authors’ own recent paper (cited above) have shown relationships on par with, and stronger than, the associations shown here. This does not seem to be a very strong motivation for the study questions investigated in this paper. Perhaps it would be better to write about how the relationship of expression of risk genes is complicated and poorly understood. In addition, the way this sentence is phrased seems to be directly contradicted a few lines later (lines 24-26).

12) “neuroinflammation are widely acknowledged as pivotal mediators of both t and amyloid30 b (Ab) pathophysiology”. “Mediates” is a rather strong word here. Many of the studies show temporal or spatial associations, but causality here is far from established. I recommend the word mediated is changed.

13) Lines 245-247, the authors write about different “strains” of tau, when referring to phosphorylation of different binding sites. These are entirely different phenomena and these sentences don’t make much sense.

14) As pointed out, this paper has many limitations and makes many assumptions. There is no limitations section enumerating these, which could lead to misunderstanding by uninitiated or less expert readers.

FURTHER MINOR COMMENTS

* It would help if Table S3 also listed how tau is quantified in each case (both the staining method and the method of quantification — the one actually shown seems to be an ordinal scoring system), as well as the time between injection (or birth in the case of non-injection) and final timepoint used in the analyses.

* In section 2.5, how were the genes chosen? It is not listed in the methods, and many don’t have any known direct relationships with tau.

* Section 2.4 should also say “are not determined by gene expression *OR spatial information”, since neither embedding showed strong relationships with tau pathology.

* Line 504: A brief explanation would be useful to summarize how these subclasses were chosen.

Version 1:

Reviewer comments:

Reviewer #1

(Remarks to the Author)

The authors have adequately addressed all concerns. No further comment.

Reviewer #2

(Remarks to the Author)

The authors have responded to the Reviewer comments admirably. I remain enthusiastic about the strengths of this paper. The addition of a limitations section, and far more measured and accurate reporting have also substantially improved the paper’s suitability for publication. Meanwhile, the additions suggested by the Reviewers have improved the paper’s quality. I have now just a few unresolved issues (relating directly to previous comments) that, despite the paper’s clear improvement, are important issues that remain to be addressed.

1) The manuscript text now does a much better job of reflecting the actual data therein, and the null modeling showcases the modest effect sizes. The authors do a very good job of discussing this. However, there remain a few prominent overstatements in the most important places: the title and the abstract.

In the abstract, the authors write:

“we have demonstrated that regional cell-type composition is a compelling explanation for the selective vulnerability observed in tauopathic diseases at a whole-brain level”.

The data do not support cell types as an “explanation” of selective vulnerability. Please adjust the abstract text to reflect the fact that, while there are some associations between cell type and tau vulnerability, those associations are modest.

Similarly, and as Reviewer 1 pointed out, the title makes an overstatement, as these modest associations are hardly “underpinnings” of tau vulnerability (at least according to this data). There is also no indication of the fact that these results are all in mouse models, which the authors also admit produce results different from those published in humans. I would recommend something like:

Cellular associations with selective vulnerability to tauopathic insults in Alzheimer’s disease mouse models

or

Searching for cellular underpinnings of the selective vulnerability to tauopathic insults in Alzheimer’s disease mouse models

or something similar.

2) In recommending the authors cross-validate their models, I previously mentioned two valid approaches — a leave-one-dataset out approach, or a k-fold cross-validation. The authors instead choose a leave-one-observation-out approach, which is not valid. This approach is no longer used in the scientific community, as it is consistently found to be optimistically biased (see e.g. Varoquaux et al. 2017 Neuroimage; Poldrack et al. 2019 JAMA Psychiatry), which is the same problem the authors started with to begin with. If the authors wish to obtain realistic estimates of how much a linear combination of cell-types can *actually* explain tau patterns, please use one of the suggested approaches.

3) The null modeling is an excellent addition to the paper and gives a much more informative estimate of effect sizes, also strengthening the author’s findings. However, they still do not address the question of whether these cell type distributions are really explaining tau patterns more than anything with a similar spatial distribution (as both reviewers brought up). In the rebuttal, the authors explained that there are no such spatially-aware null models for mouse data, but techniques like Brainsmash (<https://brainsmash.readthedocs.io/en/latest/>) can work on arbitrary volumes using just coordinates and values. This should be sufficient for the AMBA mapped data. This analysis is critical to understand whether the authors are learning cell-type relationships to tau, or are just learning the regional distribution of tau, which shares a lot of information but is not interesting. It would help to provide null models for Figure S9 as well.

Reviewer #1 (Remarks to the Author):

Torok and colleagues aimed to assess the regional association between cellular subtypes and susceptibility to tau pathology in the brain of PS19 mouse models. Cellular subtypes were identified based on single cell transcriptomics data in wild type mice provided by the Allen Institute of Brain Sciences. The mapping of tau pathology was done based on an already existing data set from multiple experimental studies on tau pathology in PS19 mice. 42 cell types identified previously by Yao et al. were mapped throughout the brain based on voxel-based MISS analyses and associated with tau pathology at the ROI level of the mouse experiments. The authors report positive and inverse association between tau and these various cell types broadly categorized according to glutamatergic, GABAergic, and non-neuronal cell types, and interpreted the findings in terms of vulnerability vs resilience. Furthermore, the authors test associations between the regional expression of AD risk genes and tau for comparison with that of the cell-types, and showed that cell-type specific correlations sometimes outperformed the AD risk genes patterns as a correlated of regional tau.

Overall, the manuscript is very well written and adds significantly to the current literature. The rationale is clear and given that the question which cellular substrates are associated with vulnerability and resilience with respect to tau pathology is still a hotly debated, the current study is timely and important. The stream of analyses which build upon each other is appealing, and framing the cell types along major axis of glutamatergic, GABAergic, and glial categories of cell types provides a useful framework. Yes, some shortcomings may be addressed as well.

Comment 1.1

(Introduction) The authors claim that the spatial expression of AD-risk genes in the brain shows little resemblance to the regional patterns of tau deposition in patients with AD. Yet, a number of studies have reported such resemblances and - beyond AD risk genes - explored regional associations between gene expression and susceptibility to tau and Abeta based on transcriptomic data from the Allen Human Brain Atlas. The authors may provide a more balanced overview of the findings (e.g. from the Sepulcre lab, who studied that in depth).

We appreciate this comment from the reviewer, and we have added some additional context in the **Introduction** as follows:

There exists a notable dissociation between where **upstream AD risk** genes are normally located in the brain and downstream pathology, an observation that has been called one of the key mysteries in the field of neurodegenerative diseases [Fusco 1999, *J Neuro*; Subramaniam 2019, *The Yale Journal of Biology and Medicine*]. **Taking a broader perspective on the interactions between gene expression and the development of AD-associated pathology may be required; for instance, Sepulcre, et al. examined propagation patterns of tau and amyloid- β (A β) and found associations for hundreds of genes previously unknown to be associated with AD, utilizing transcriptomic data from the Allen Human Brain Atlas [Hawrylycz 2012, *Nature*; Sepulcre 2018, *Nature Medicine*]. ~~A partial explanation of both selective regional vulnerability and its dissociation with associated genes is potentially~~ Furthermore, the dissociation between AD risk gene**

expression and selective regional vulnerability may be partly due to the fact that certain cell types, especially subtypes of glutamatergic neurons, in affected regions harbor significantly more τ -inclusions and degenerate at a faster rate than others.

Comment 1.2

(Introduction) A strength is that data from different experimental methods to induce tau pathology in PS19 mice were taken into account to avoid confounding of regional differences in tau pathology as a result of technical idiosyncracies. Having said that, it is important to point out that tau pathology is not a uniform type of brain alteration, but systematic differences in the form of tau isoforms/strains exist that may result in marked differences in terms of cell types, distribution pattern, and phenotypes. For example, in one of the data sets from PS19 mice the authors included (Boluda et al. Acta Neuropathol 2015), tau pathology from AD patients or CBD patients were injected in PS19 mice, resulting in strong differences of tau pathology by cell type, tau spreading etc. Yet, both experimental conditions were pooled in the current study. Overall, pooling data across such meaningful differences may have watered down any cell-type specific regional differences in vulnerability to AD-like tau pathology.

The authors may better explain their rationale for the methodological approach they took. As mentioned, there are certainly advantages in the current approach, but the disadvantages need at least to be discussed as a caveat.

We thank you reviewer for an encouraging and highly constructive critique. We fully agree that the heterogeneity between tau experiments is both a strength and a limitation of the present study. By using a meta-analytical approach, we were able to assess *general features* of tauopathy across studies with these heterogeneous experimental conditions; this is why we make statistical associations per study, and only aggregate the resulting statistics to glean generalizable associations and effects (e.g., **Figures 2** and **4**). To further highlight this point, we have added **Table S7**, which lists the names of the top-5 most correlated cell types for each study.

To address the concern that we have not adequately addressed the drawbacks of our approach, we have added the following to a new **Limitations** subsection of the **Discussion (3.3)**:

Secondly, to help minimize the numbers of different features between datasets, we relied on studies that used mice with the PS19 background (**Table S3**). However, while this is a widely used model of tauopathy, the P301S tau mutation that it carries is not characteristic of AD, even if the injectates used may be derived from AD subjects (e.g., DS4 from Kaufman, *et al.* [Kaufman 2016, *Neuron*]). Further, our decision to pool these 12 datasets together allowed us to meta-analytically evaluate which cell-type features may underpin tauopathy more generally, but also introduced significant variability into our results that may have obscured more nuanced relationships.

Comment 1.3

Could the authors please specify for the different tau data sets which measures of tau were used (i.e. AT8 antibody, PhF staining etc).?

We have modified **Table S3** accordingly.

Comment 1.4

The authors reported the Pearson Moment correlations between gene expression and tau pathology (Figure 2). The cell-type dependent gene expression in the mouse brain may however be highly restricted to certain brain regions (Yao et al, Nature 2023, Yao et al Cell, 2021), entailing potentially a non-normal distribution of gene expression when analyzed across regions. Could the authors report the distribution properties and confirm that Pearson-moment correlations rather than alternatives such as Spearman correlation are appropriate?

This is an excellent point. We note that all correlations in the paper are assessed *only in regions where tau was quantified* for each study, which covers variable numbers of regions depending on the study (**Table S3**); therefore, we expect that this issue is not so severe. However, we have assembled complementary figure panels for the Supplement (new **Figure S9**) showing the results for Spearman’s correlation, with qualitatively similar results (see below). Notably, CA1-ProS and oligodendrocytes remained the most highly positively and negatively correlated cell types, respectively.

Comment 1.5

The authors mention that the correlation between CA1-ProS expression and tau stood out as the single most vulnerable cell type which was highly confined to the CA1 region. (line 108). However, a visual inspection of the scatter plots suggest that other cell types of this hippocampal glutamatergic class showed a similar size of correlation, at least one that may not be significantly different from that of the CA1-ProS. Could the authors please support their stance by statistics?

This is an excellent point. While we drew particular attention to this cell type in the **Discussion (3.1)**, many other cell types did not have significantly different mean correlations from CA1-ProS (2-sample t-test, $p > 0.05$), particularly the other hippocampal glutamatergic neurons. Ultimately, we did not include this analysis in our results because, with only 12 studies and 42 cell types, we lacked the statistical power to make a firm assertion about the pairwise differences between cell types in mean correlation; instead, we focused on pairwise differences between classes of cell types. Although we do not believe this issue confounds the general interpretation of our results – that hippocampal glutamatergic neurons as a class are the most correlated to tau pathology (**Figure 2B**) – we have made clear in the text that CA1-ProS is not *uniquely* strongly correlated to tau pathology across studies. See also the response to **Comment 1.8** below, where we address the issues of low correlations in general.

We do note, however, that CA1-ProS is among the several cell types whose correlation to tau pathology was statistically significant; see **Comment 1.6** below.

Comment 1.6

Did the authors correct for spatial auto-correlation?

While we agree with the reviewer that spatial autocorrelation is an important consideration, we did not directly correct for it, as there were no readily available methods for performing spin permutations for the mouse brain. However, to partly address this issue, we constructed distributions of null models where the regional values of each cell type were first scrambled prior to performing associations with tau pathology. Below we show the distributions of mean Pearson's R value across datasets for each collection of null models ($n = 10,000$ per cell type) alongside the true mean R for that cell type; this has now been incorporated into **Figure 2A**:

After Bonferroni correction, the cell types whose mean Pearson's R was statistically significantly different from those of the null models ($p < 0.05$) were: CA1-ProS, CA3, Meis2, Micro-PVM, Oligo, SUB-ProS, and Sst-Chodl.

Comment 1.7

The result showing a change in directionality of the correlation coefficient between glutamatergic subtypes of the hippocampus vs isocortex was surprising. Possibly a GO analysis to interrogate the biological pathways distinguishing the effect of glutamatergic subtypes between both brain regions is useful. It is up to the authors to consider such an analysis.

This is an interesting analysis that we had not previously considered. Using the same measure of differential expression as with the **Figure 7** GO analysis where we specifically looked at the glutamatergic neuron subtypes (see **Methods (5.6)**), we obtained the 100 most differentially expressed genes as below; now **Figure S22**). We also acknowledge that it is possible that the genes identified between these two sets of cell types may be poorly annotated for central nervous system functions. We cross each of the cortical and hippocampal classes and ran GO analysis. While the processes were indeed distinct between these neuronal classes, only the hippocampal glutamatergic neurons specifically expressed genes that were associated with neuronal functions, particularly with respect to the regulation of GABAergic synapses. While preliminary – and with the noted caveat that the annotation of these genes in the CNS is likely incomplete – this is an intriguing result, since the hippocampal glutamatergic neurons positively associated with tau (**Figure 2B**).

Comment 1.8

(Discussion) Overall the correlation coefficients are relatively small, in particular given the substantial variability in the size of correlation coefficients depending on the tau mouse model chosen. The authors may discuss that point.

Both R1 and R2 have raised this point. The correlations are indeed moderate; we observed no single average R above 0.4 for any of the explored cell types or genes, and no single R for any study above 0.7. This should be expected, since we do not believe regional vulnerability can be fully or even mainly explained by cell-autonomous or region-specific factors. In fact there is burgeoning literature in support of non-cell-autonomous factors like trans-neuronal tau transmission, and the role of neuroinflammation. Hence it was not our goal, nor is it plausible, to detect overly strong cell type associations with tau. Instead, our goal was to assess all evidence relating cell distributions to tau, so that their relative contribution to tau pathophysiology may be quantified in an unbiased manner. We believe our presented results support this contention rather well. Nonetheless, we note that certain cell types do stand out in the strength of their correlation, such as oligodendrocytes.

We have added the following text to the **Discussion (3.1)** to address this point:

Below we discuss several notable individual cell types in light of their contributions to SV-C or SR-C. These are broad findings that were found to generalize across studies. We emphasize that a high degree of variability exists between individual cell types within each class and between datasets (**Figure 2A** and **2B**). No single correlation between any dataset's end-timepoint tau pathology and any cell type was higher than 0.7 in magnitude, nor were any mean correlations across datasets greater than 0.4. This That the associations were modest-to-moderate in strength suggests that a more nuanced understanding of the cellular underpinnings of tau vulnerability would require detailed mechanistic investigation of specific cell types identified here [Leng 2021, *Nature Neuroscience*; Gabitto 2024, *Nature Neuroscience*]. We further note that an assessment of the significance of pairwise differences between cell types could only be performed at the class level (**Figure 2B**) because of the small number of datasets relative to the number of individual types. Nevertheless, we make mention of cell types that stood out both in the univariate (**Figure 2**) and multivariate (**Figure 5**) analyses.

Comment 1.9

(Discussion) Please discuss differences between mouse and human gene expression. Several studies did report conserved spatial gene expression patterns between mice and humans, but also that gene expression of cell types involved in the immune response, i.e. a key factor in tau progression, may differ (e.g. Zhou et al. Nature Med 2020; Miller et al. PNAS 2010).

The reviewer makes an important point, which we have noted in the new **Limitations** subsection of the **Discussion (3.3)**:

We also note that gene expression in mice differs from that in humans in important ways, some of which bear relevance to AD [Miller 2010, *PNAS*]; in part this may relate to our results concerning the *Rorb* gene (**Figure 4A**). However, we note that there is notable homology between cortical cell types in mice and humans despite divergent evolutionary histories [Hodge 2019, *Nature*], and gene-expression-related changes in AD have also been observed in mouse models [Zhou 2020, *Nature Medicine*].

Comment 1.10

(Title) Given that tau pathology was assessed in the PS19 mice which carry the MAPT mutation which do not fully recapitulate the type of 4R/3R tau pathology of AD, the claim to have uncovered the cellular underpinnings of vulnerability to tau in Alzheimer's disease seems a bit strong and the title may be toned down.

Among mouse models of tauopathy, PS19 background mice are the most commonly used. Although the reviewer correctly notes that the P301S mutation is not characteristic of AD tau, nor do these mice exhibit the mix of 3R/4R isoforms observed in clinical AD subjects, previous work has utilized PS19 mice to explore features of AD pathophysiology. A key determinant of tauopathy spread is the *injected seed*, which differs greatly between studies (**Table S3**). For instance, Hurtado et al. explored a bigenic amyloid/tau mouse model [Hurtado 2010, *The American Journal of Pathology*], and both Kaufman et al. and Boluda et al. seeded their PS19 mice with AD-derived tau [Kaufman 2016, *Neuron*; Boluda 2015, *Acta Neuropathologica*]. These studies found spread

patterns that were distinct from non-AD tauopathy models and exhibited features that mimicked human AD.

We wholeheartedly agree that no transgenic model can ultimately fully reproduce a largely sporadic and multifactorial disease like AD; however, the value here is that the ramification of tauopathy, the most critical pathological hallmark of human disease, is abundantly present and amenable to statistical modeling. We believe the title therefore suits the present work. Nonetheless, we have dialed down our claims (see the response to **Comment 1.8**), and are now more careful in stating wherever possible that these results pertain to transgenic models of AD-related tauopathy. We have also commented on these potential issues in the new **Limitations** subsection of the **Discussion (3.3)**.

Reviewer #2 (Remarks to the Author):

Torok et al. describe an analysis to investigate regional vulnerability to tau pathology by looking at atlas-based colocalization between tau, cell-type distribution and gene expression patterns in the mouse brain (specifically cortex and subcortex). The authors show that the distribution of certain cell-types and the expression patterns of certain genes show modest correlations with the distribution of tau in the mouse brain. While these findings are not validated against human single-cell atlas data, several of the findings conflict with those of such studies. The authors build inferential models trying to explain the distribution of tau as a combination of cell-types (or top genes) and, in doing so, conclude that cell-type distribution better explain tau accumulation patterns than gene expression patterns. However, the authors also show that cell- or gene-based spatial models of tau distribution seem to differ depending on at what time point during disease progression the models are formed.

The major strengths of this paper include a very creative curation, re-use and combination of datasets. The description of this data, and the degree to which it is shared is a great strength. Finally, the research question targeted by this paper is timely and of great interest to the AD field. While this paper features very interesting data, the interpretation of this data in the manuscript seems problematic. Several of the analyses contain important conceptual or methodological flaws, many large assumptions are made without being acknowledged, and at times it feels like a narrative is being pushed (in abstract and throughout manuscript) that isn't well supported by the data. These issues are expanded at length below:

Major Comments:

Comment 2.1

(Conceptual) The authors show how the distribution of cell types across regions relates to tau. But how do we know that this says something about cellular vulnerability? None of the relationships are very strong in the paper, meaning specific cell types don't seem to very strongly mediate regional vulnerability. Just the fact that cell type varies by region indicates that any regional pattern should show cell-type relationships, even if weak (see null modeling point, #8 below). For example, tau is known to accumulate less in heavily myelinated areas, and heavily myelinated areas have more oligodendrocytes. The authors replicate these findings, but try to push a story about oligos mediating tau resistance, citing somewhat obscure mechanisms and papers. This association could just as likely (perhaps more likely) have to do with myelination or factors that drive it than it does with a direct role of oligos in fending off tauopathy. In fact, oligodendrocytes themselves accumulate tau pathology in 4R tauopathies like PSP, which seems to suggest against the idea that they have a natural resistance to tauopathy. This point is driven even further by the fact that linear combinations (that are not cross-validated, see point #9 below) are necessary to find strong associations in this paper — it is very likely that these models are simply learning regional patterns, not regional vulnerability. I understand that inference is limited from spatial associations, but we are talking about mouse models where direct experiments are possible at a single-cell level. Therefore, it is hard to understand why spatial association analyses in mice are needed (and interesting how many of the results seem to contradict scRNA human

findings). If nothing else, this should be listed as a major limitation of the paper (there isn't even a limitations section as of now), and throughout the paper the authors really need to remove the various strongly worded sentences about mechanisms and assumptions of directionality or any kind of molecular interaction, which is not supported.

We appreciate these concerns from the reviewer and have addressed them in several ways.

First, we agree that our study is limited to studying *innate* regional vulnerability/resilience, as we can only compare healthy gene expression and cell-type distributions with tau deposition in the disease condition. Similarly, we cannot and do not attempt to probe at mechanistic relationships between cell types and tau. However, we believe that, although this narrows the scope of our analyses, exploring questions of innate SV/SR in disease is highly clinically relevant and follows in the footsteps of previous analyses connecting the baseline gene expression to proteinopathy (see, for instance, [Acosta 2018, *Alzheimer's & Dementia*; Anand 2022, *Scientific Reports*; Henderson 2019, *Nature Neuroscience*; Dadgar-Kiani 2022, *Cell Reports*]). We anticipate that future experimental work will be able to jointly probe single-cell-level gene expression and tau vulnerability at a *whole-brain level*, but at present, no such resource exists. For instance, the state-of-the-art SEA-AD resource from the AIBS only contains data from the middle temporal gyrus [Gabbito 2024, *Nature Neuroscience*]. This study sought to complement that work by looking more broadly at why certain regions are relatively more affected by tau pathology than others.

The scope of the study has been clarified in the **Introduction** as follows:

This computational technique enabled us to significantly expand the number of covered cell types and their corresponding spatial coverage beyond those for cell types obtained by current *in situ* sequencing methods. These inferred cell-type densities could then be compared with regional distributions of tau quantified in mouse models of tauopathy, allowing us to probe questions of how the presence of cell types in a given region confers vulnerability or resilience to the later development of tau pathology; that is, *innate* SV-C and SV-R.

As the reviewer rightly suggests, however, we have included this as a point within the new **Limitations** subsection of the **Discussion (3.3)**:

A further set of limitations stems from the fact that we were restricted to comparing *baseline* cell-type densities – that is, those in the healthy mouse brain – with tau distributions in the disease condition. While we expect that cell-type densities are more stable and robust to disease-associated perturbations relative to gene expression, some cell types, such as microglia, have dynamic populations. Therefore, our claims can only be interpreted through the lens of *innate* regional vulnerability to tau. As emerging technologies facilitate the study of disease-associated changes in gene expression, cell-type distributions, and tau deposition at the whole-brain level in the same mice (or humans presenting with tauopathy), we can more deeply delve into questions of *dynamic* regional vulnerability.

We have also addressed the valuable points about the correlation strength of individual cell types, the construction of appropriate spatial nulls, and cross-validation; see the responses to **Comments 2.2, 2.8, and 2.9**.

Comment 2.2

*(Conceptual) Similarly, throughout the paper, there is language suggesting that strong associations are seen. Some examples (my *emphasis* added):*

*Line 113-114: By contrast, the immune cell subtype comprising microglia and perivascular macrophages (Micro-PVM) exhibited a *strong positive correlation*.*

*Line 143: “Trem2, a marker for disease-associated microglia (DAM), also generally *aligned closely* with overall t distributions”*

This is quite deceptive language, given that there are no strong associations suggestive of actual mechanistic interaction in any of the univariate analyses in the paper. This language must be changed.

We agree that the strength of these correlations is overstated and have amended those lines and others accordingly; see for instance:

By contrast, the immune cell subtype comprising microglia and perivascular macrophages (Micro-PVM) exhibited a **strong** positive correlation.

Trem2, a marker for disease-associated microglia (DAM), also generally aligned **closely** with overall tau distributions.

In addition to softening our language, we have added the following text to the **Discussion (3.1)**:

Below we discuss several notable individual cell types in light of their contributions to SV-C or SR-C. These are broad findings that were found to generalize across studies. We emphasize that a high degree of variability exists between individual cell types within each class and between datasets (**Figure 2A and 2B**). **No single correlation between any dataset's end-timepoint tau pathology and any cell type was higher than 0.7 in magnitude, nor were any mean correlations across datasets greater than 0.4.** **This** That the associations were modest-to-moderate in strength suggests that a more nuanced understanding of the cellular underpinnings of tau vulnerability would require detailed mechanistic investigation of specific cell types identified here [Leng 2021, *Nature Neuroscience*; Gabitto 2024, *Nature Neuroscience*]. **We further note that an assessment of the significance of pairwise differences between cell types could only be performed at the class level (Figure 2B) because of the small number of datasets relative to the number of individual types. Nevertheless, we make mention of cell types that stood out both in the univariate (Figure 2) and multivariate (Figure 5) analyses.**

We do note, however, that several of the individual cell types, including CA1-ProS and Oligo, exhibited statistically significant associations across datasets as compared with naive spatial null models (see the response to **Comment 2.8**).

Comment 2.3

(Conceptual) Are the authors concerned that the mouse models used have a P301L background, a mutation which causes 4R tauopathies, which is a class of disease that does not include AD? Could this have something to do with the oligodendrocyte findings, since oligodendrocyte tau is a phenotype observed in 4R tauopathies, but not AD?

As the reviewer correctly notes, all mice in this study have a mutation that is not associated with AD (although it is P301S, not P301L). We have mentioned this concern in the new **Limitations** subsection of the **Discussion (3.3)**:

Secondly, to help minimize the numbers of different features between datasets, we relied on studies that used mice with the PS19 background (**Table S3**). However, while this is a widely used model of tauopathy, the P301S tau mutation that it carries is not characteristic of AD, even if the injectates used may be derived from AD subjects (e.g., DS4 from Kaufman, *et al.* [Kaufman 2016, *Neuron*]). Further, our decision to pool these 12 datasets together allowed us to meta-analytically evaluate which cell-type features may underpin tauopathy more generally, but also introduced significant variability into our results that may have obscured more nuanced relationships.

On the specific point about oligodendrocytes, we note that there are several recent studies ([Gabbito 2024, *Nature Neuroscience*; Green 2024, *Nature*; Mathys 2023, *Cell*]) that have noted AD-associated changes in oligodendrocyte subpopulations. We also do not and cannot claim that the negative association between oligodendrocytes and tau pathology at the whole-brain level is directly related to tau internalization, or any other mechanism. In the **Discussion (3.1)**, we postulate that it may reflect that homeostatic functions carried out by oligodendrocytes, such as the maintenance of myelin, are affected by tau, and those regions with lower pools of oligodendrocytes may therefore be less resilient to tau pathology. However, this is merely a hypothesis meant to be explored by future work.

Comment 2.4

(Conceptual) The comparison between cell types and individual genes in describing the distribution of tau produces one of the main findings described in the abstract. However, this analysis has several important flaws that cast doubt as to the authors' conclusions. First, the highest spatial correlations between tau and top AD-associated genes are equivalent to the associations with cell-type (median around 0.36), making the claim that one is superior hard to understand. Second, from a statistical perspective, the authors are not comparing apples to apples. Single transcriptomic measurements are notoriously unstable (both methodologically and biologically) and often don't replicate particularly well. However, transcriptomic networks tend to be more robust, replicable and predictive, and have a better SNR. Cell-types, even more organized, composed of many transcriptomic networks, and have even better SNR. Are the authors sure the failure of single genes to outperform cell-types in the Figure 5 analysis not just driven by the different SNR of these two measures? What if the authors compared cell-types to transcriptomic networks (generated e.g. through WGCNA or equivalent)? This would at least be a slightly fairer comparison.

While SNR is an uncontrollable factor that we inherit from the methodologies through which the gene expression data were collected, we note that that AGEA has been widely utilized at the single-gene level and is the most comprehensive spatial atlas of gene expression currently available. Both our lab and others have successfully used this resource to probe questions of innate vulnerability in the context of proteinopathic disease (e.g., as mentioned above, [Acosta 2018, *Alzheimer's & Dementia*; Anand 2022, *Scientific Reports*; Henderson 2019, *Nature Neuroscience*; Dadgar-Kiani 2022, *Cell Reports*]). We do agree with the reviewer that cell types should possess better SNR than individual genes; indeed, this was part of our motivation for using cell types in our study design.

In the spirit of using a more robust measurement, we constructed an *AD risk eigengene*, which we took as the first principal component of the 24 AD risk genes we included in our study (which came from the Alzheimer's Disease Sequencing Project; see the response to **Comment 2.16**). We have included this as a correlate in **Figure 4A** (see below). It exhibits a net-positive correlation ($R = 0.27$), but also a high degree of variability, and its association with tau does not stand out among the individual genes.

While it does not get at the question of SNR directly, we also note that with the new leave-one-out cross validation results that we obtained (see the response to **Comment 2.9**), our findings vis-à-vis cell types outperforming AD risk genes were reinforced.

Comment 2.5

(Conceptual) The authors interestingly find that RORB distribution does not associate with tau pathology, contradicting actual single-cell work. However, the author themselves has recently published a study using a similar analysis in human data that found a positive association with RORB and tau distributions (10.1101/2024.03.04.583403). How do the authors reconcile these conflicting findings, or the ability of this approach in mice to be meaningful in humans?

We believe that this discrepancy may be reconciled by noting that *RORB*-expressing glutamatergic neurons are found throughout layer-4 of the cortex, and not simply in the entorhinal cortex, where they are expected to contribute to *primary* neuronal vulnerability (e.g., [Leng 2021, *Nature Neuroscience*]). Since end-timepoint pathology in these mouse datasets is *relatively*

cortex-sparing (with noted variability), we did not find the positive association between tau and mouse *Rorb* expression suggested by previous work with AD-derived neurons and our own past work in clinical AD data, which, by comparison, exhibits pronounced cortical pathology. This is an important limitation of our study, which we discuss in the new **Limitations** subsection of the **Discussion (3.3)**.

On the specific point about *RORB*, we have also added the following to the **Discussion (3.1)**:

Neurons isolated from entorhinal regions, however, largely did not ~~confer vulnerability to~~ **positively associate with** tau across the 12 datasets (**Figure 2A**) nor did they feature in the multivariate models (**Figure 5B** and **S10B**), despite the fact that the EC is one of the earliest regions to exhibit tau pathology in AD [Braak 1991, *Acta Neuropathologica*]. By contrast, scRNAseq performed on postmortem AD patients identified glutamatergic neurons in layer II of the entorhinal cortex [Fu 2018, *Nature Neuroscience*], and in particular those expressing the gene *RORB* [Leng 2021, *Nature Neuroscience*], to be especially susceptible to tau. **We also did not find a notable positive association between the baseline expression of *Rorb* in the mouse brain and tau pathology (Figure 4)**; likely, this is due to the fact that *Rorb* is also a marker of L4 glutamatergic neurons in the cortex and tau pathology in these datasets is lower in cortical areas than in limbic structures (**Figure S7**). **Intriguingly** However, the unseeded Hurtado *et al.* mouse model revealed pronounced early EC pathology (**Figure 6B**), which did exhibit strong correspondence to EC-isolated excitatory neurons such as L3 IT ENT (**Figure 2A** and **6D**) [Hurtado 2010, *The American Journal of Pathology*]. **Overall, this may reflect the inherent limitations of the present work to explore primary cellular vulnerability, by the nature of the datasets that are currently available (see Limitations below).** ~~Therefore,~~ it may be necessary to use mouse models that develop tau endogenously, more closely mimicking human disease conditions, in order to better study primary selective vulnerability [Fu 2019, *Nature Neuroscience*] to tau pathology.

Comment 2.6

(Conceptual) It would help for the authors to make explicit how using the final timepoint affects interpretations. By using the final timepoint, the level of tau is likely somewhat saturated. This means the authors are likely assessing which regions simply do and do not (ever) get tau. This is a perfectly fine way to measure vulnerability/resilience, but it would differ from e.g. measuring vulnerability as tau arrival time. Certain regions may have lower “ceilings” of pathology (e.g. the entorhinal in cortex in humans, which is perhaps the most vulnerable but actually has relatively low levels of tau at saturation). Would the authors expect to see different results if they instead profiled vulnerability as arrival time over time, and what would that say as a measure of vulnerability/resilience? This is partially addressed in Section 2.7, but here the authors seem to change the definition of vulnerability/resilience, which confounds the discussion and interpretation the findings. If the authors trained the model on the final time point and used it to predict earlier timepoints, would it work? If not, what does that really say about this Method’s ability to parse cell-based vulnerability/resilience? This section is, by the way, not covered sufficiently in the Methods to really understand what the authors have done.

We appreciate the reviewer’s point about tau saturation and the confusion over the definitions of ‘vulnerability’ and ‘resilience’. On the latter point, we have made a distinction between *primary* (propensity to endogenously produce pathological tau) and *secondary* (propensity to be affected by tau spreading from other areas), as outlined by Fu et al. [Fu 2019, *Nature Neuroscience*]. We emphasize that, outside of the analysis for the Hurtado 2010 dataset, we cannot assess primary vulnerability to tau, as early timepoints are confounded by the fact that a large quantity of pathology has been introduced at the seeding site. Thus, most of our analyses concern secondary vulnerability, which is why we chose to model end-timepoint pathology instead, removing the seeded region prior to fitting any of our models to further mitigate seeding effects (see **Methods (5.3)**).

Though it does not represent true primary vulnerability, to more thoroughly explore the question, we show the results of our univariate and multivariate modeling when predicting either timepoint 1 (first set of panels) or timepoint 2 (second set of panels) below. For the univariate, both largely recapitulate the results shown in **Figures 2**, which is not unexpected given that there is substantial inter-correlation between timepoints per study (**Figure S8**). These results have been shown in new **Figure S10**.

Further, we found that, although the cell-type models all fit the data at a significance of $p < 0.001$ or less for timepoint 1 (**A**), there were some notable differences between the frequency of several notable cell types selected for this timepoint (**B**) relative to timepoint 2 (**D**) or the last timepoint (**Figure 5B**). Most prominently, Micro-PVM is one of the two highest-frequency cell types, and both CA1-ProS and Oligo – the two most significantly correlated types for end-timepoint pathology – appear less frequently, with Oligo not being selected for any of the 12 datasets. Though we cannot rule out the unintended effects of seeding (even though the seed region is not included in any of these analyses), this agrees with the result found for the Hurtado 2010 dataset (**Figure 6A**), where the linear models only included Oligo at the last 2 of the 4 timepoints (modified, renumbered **Table S9**). These results have been included as new **Figure S20**.

On the final point, we have also clarified the linear modeling procedure in **Method Subsection 5.5** as follows:

All analyses were performed using MATLAB v.2022³. ~~For the linear modeling, we first performed feature selection to minimize the risk of overfitting.~~ Unless otherwise stated – see, for instance, **Results (2.7)** – linear models were fit to end-timepoint tau pathology (**Figure S7**) for each study individually using the MATLAB `fitlm` function. For the linear models using the distributions of multiple cell types or AD risk genes, we first performed feature selection to reduce the risk of overfitting.

and:

As mentioned above, an intercept term was included for all models. Both procedures yielded qualitatively similar results in terms of model fits (through R^2), statistical significance, and BIC values. We also performed leave-one-out (LOO) cross-validation for the BIC-selected cell-type and AD-gene models using the MATLAB `fitrlinear` function, from which F -statistics and log-likelihoods could be calculated to determine R^2 , p , and BIC values.

We have made clear what we mean by vulnerability, distinguishing between primary and secondary, throughout the text, and also address the shortcomings of mouse models in the new **Limitations** subsection of the **Discussion (3.3)**.

Comment 2.7

(Conceptual) In section 2.8, the authors appear to be surprised that genes underlying “vulnerable cell types” differ from AD risk genes associated with tau. However, top expressed genes in cell types likely have to do with cell identity and maintenance. If those genes also related to disease, you would very likely see important morphological or phenotypic differences in certain cell types, which is not a characteristic of AD. I don’t fully understand the purpose behind this analysis, but the results are expected.

The reviewer is right that this finding is not entirely unexpected. We find it noteworthy nevertheless, as it underscores the fact that cell-type-mediated vulnerability/resilience (SV-C/SR-C) is distinct from single-gene-mediated vulnerability/resilience (SV-G/SR-G), with the caveat that both of these are measured in the baseline, healthy condition.

Comment 2.8

(Methodology) The ability to assess the findings described in this manuscript are greatly impeded by the fact that the underlying data is often not shown. First, it would be very helpful to show the regional (final timepoint) spatial maps of tau for the various mouse-model datasets. This will give a good sense of how variable these datasets are. Importantly, we do not know the distribution of these datasets and whether regular parametric comparisons are appropriate. Related to this point, it would be very helpful to see scatter plots (similar to Figure 5A) for the important associations in Figure 2 and 4. Visualizing them all is obviously not necessary, but perhaps the top associations.

This will be helpful for establishing whether some of the associations are driven by just a few regions (which I suspect for at least some of these univariate associations, especially the finding of hippocampal neurons being most associated with pathology in Figure 2C). The nature of these points stems from concern over correlating maps with strong spatial features. It is important that the authors use spatially-preserving null models (10.1016/j.neuroimage.2021.118052) to establish how strong an association for each of these features would be expected by chance. In most cases, it will likely be non-zero.

While too extensive for the main text, we have provided the distributions of the final timepoint distributions of tau for the 12 datasets used here in **Figure S7**, and we believe the differences in tau pathology between experimental conditions are striking. Scatter plots for the highest correlation and anti-correlation across cell types and studies are shown below and have been incorporated into a revised version of **Figure 2C** (see below). Although we did not directly correct for it, as there were no readily available methods for performing spin permutations for the mouse brain, we agree that spatial autocorrelation is a potential concern. However, to partly address this issue, we constructed distributions of null models where the regional values of each cell type were first scrambled prior to performing associations with tau pathology. Below we show the distributions of mean Pearson's R value across datasets for each collection of null models (n = 10,000 per cell type) alongside the true mean R for that cell type; this has now been incorporated into **Figure 2A**:

After Bonferroni correction, the cell types whose mean Pearson's R was statistically significantly different from those of the null models ($p < 0.05$) were: CA1-ProS, CA3, Meis2, Micro-PVM, Oligo, SUB-ProS, and Sst-Chodl.

We have also repeated the analyses in **Figures 2A** and **2B** using Spearman's rho rather than Pearson's R, and found qualitatively very similar results (new **Figure S9**):

Comment 2.9

(Methodology) Section 2.6 involves creating linear combinations of cell-types in order to predict the distribution of tau. The authors make a few statements about protection from overfitting, but BIC is unfortunately not sufficient to assess whether overfitting is happening. The real pursuit here is whether the presence of a cell-type can predict where tau accumulates. The best way to do this is through cross-validation — if one builds a model of cell-type vulnerability to tau, can it predict the distribution of tau in different data based on cell-type distribution? There are two ways the authors can do this. The first is a leave-one-dataset-out approach where the model is fit on all mouse datasets and tested on a left-out one (repeatedly) — how good is the predictive performance in such cases? The other way would be a kfold cross-validation leaving out (e.g.) 10% of regions at a time. If cell type is driving this phenomenon, a model trained on most regions should be able to predict whether tau is present in other regions. If the authors do not form this kind of cross-validated model, I am not convinced that their current models are not overfit, optimistically biased, and therefore perhaps not informative.

We first note that each linear model is study-specific; that is, the end-timepoint tau pathology of each dataset is fit independently with its own BIC-selected cell types. However, the suggestion of using leave-one-out (LOO) cross-validation approach, whereby we fit a collection of models on n

- 1 regions for each dataset, is an excellent one for increasing the robustness of our results. Below we show the LOO results for the same sets of cell-type (*top*) and AD-gene (*bottom*) predictors as in **Figure 5** and renumbered **Figure S15**, respectively. While the strength of the associations generally decreased, as expected, the cell-type models remained statistically significant (*F*-test) at a $p < 0.01$ level with the sole exception of the DS9 dataset. Further, the cell-type-based models exhibited lower BIC values than their AD-gene-based counterparts for 8/12 datasets, reinforcing the general points made in the manuscript.

Cell Type:

AD Gene:

These panels comprise new **Figures S13** and **S17**, and a summary of these results have been included in new **Table S8**.

Comment 2.10

(Methodology) One concern I have is that we don't have a great sense how accurate the cell-type decomposition is, especially in terms of specificity. The authors show a nice validation in Figure 1B, but we do not see how well these predictions predict other cell types. After all, as the authors show, the cell types have a great amount of transcriptomic overlap, and it is the transcriptomic information that the algorithm uses for decompositions. At the very least, it would be good to show a 3x3 matrix showing how predicted PVALB/SST/VIP relate to empirical PVALB/SST/VIP distribution. However, it would be ideal to use a few of the other decomposition algorithms out there and show that the results are similar. For example, why is it that some regions are not expressing astrocytes? Surely astrocytes are present in all brain regions?

The cross-correlation plots for Pvalb/Sst/Vip interneurons is an excellent addition. We have included this as new **Figure S1**; as expected, the only significant positive associations between inferred and empirical cell type distributions of the same type:

On the point about astrocytes (and others): the visualizations were thresholded for clarity, so they represent areas of highest density. Indeed, all regions contain astrocytes. This was an unfortunate omission on our part, and we have added a subsection **Visualization** to **Methods (5.7)** to clarify this point.

Comment 2.11

(Intro/Discussion) P1, Lines 12-14 — “Remarkably, however, the topography of vulnerable regions in AD appear to bear little relation to that of the factors that presumably cause it, especially expression of associated genes”. This is a curious comment. Many studies, including by the authors themselves, have shown spatial correlations between vulnerable brain regions and expression of key AD risk genes. Just as an example, several papers have shown MAPT expression to be associated with tau (10.1016/j.jalz.2017.02.011, 10.1093/brain/awy189, 10.1016/j.celrep.2024.113691), and APOE expression to be associated with tau accumulation in APOE e4 carriers (10.1212/WNL.0000000000011270, 10.1001/jamaneurol.2019.4421, 10.1126/scitranslmed.abl7646). Even the authors’ own recent paper (cited above) have shown relationships on par with, and stronger than, the associations shown here. This does not seem to be a very strong motivation for the study questions investigated in this paper. Perhaps it would be better to write about how the relationship of expression of risk genes is complicated and poorly understood. In addition, the way this sentence is phrased seems to be directly contradicted a few lines later (lines 24-26).

The reviewer's point is well taken. We have amended the language of the **Introduction** accordingly:

It has long been thought that regional vulnerability must be a consequence of the molecular composition of certain regions, which is in turn governed by upstream genes that are involved in disease pathology [Saxena 2011, *Neuron*]. ~~Remarkably, however, the topography of vulnerable regions in AD appear to bear little relation to that of the factors that presumably cause it, especially expression of associated genes [Mezias 2017; Subramaniam2019].~~ There exists a notable dissociation between where ~~upstream AD risk~~ genes are normally located in the brain and downstream pathology, an observation that has been called one of the key mysteries in the field of neurodegenerative diseases [Fusco 1999, *J Neuro*; Subramaniam 2019, *The Yale Journal of Biology and Medicine*]. Taking a broader perspective on the interactions between gene expression and the development of AD-associated pathology may be required; for instance, Sepulcre, *et al.* examined propagation patterns of tau and amyloid-beta (A-beta) and found associations for hundreds of genes previously unknown to be associated with AD, utilizing transcriptomic data from the Allen Human Brain Atlas [Hawrylycz 2012, *Nature*; Sepulcre 2018, *Nature Medicine*]. ~~A partial explanation of both selective regional vulnerability and its dissociation with associated genes is potentially~~ Furthermore, the dissociation between AD risk gene expression and selective regional vulnerability may be partly due to the fact that certain cell types, especially subtypes of glutamatergic neurons, in affected regions harbor significantly more tau inclusions and degenerate at a faster rate than others.

Comment 2.12

(Intro/Discussion) "neuroinflammation are widely acknowledged as pivotal mediators of both t and amyloid-b (Ab) pathophysiology". "Mediates" is a rather strong word here. Many of the studies show temporal or spatial associations, but causality here is far from established. I recommend the word mediated is changed.

We have removed all mentions of "mediate" in the text and changed the language as suggested; for instance, in **Results (2.8)**:

These findings underscore that the genetic and mechanistic bases of selective vulnerability and resilience differ substantially ~~when mediated by~~ **from the perspective of** cell types versus genes.

and **Discussion**:

... 4) Are SV-C and SR-C ~~mediated by~~ **related to** the same genes and functional gene networks as gene-expression-based selective vulnerability (SV-G) and resilience (SR-G)?

Comment 2.13

(Intro/Discussion) Lines 245-247, the authors write about different "strains" of tau, when referring to phosphorylation of different binding sites. These are entirely different phenomena and these sentences don't make much sense.

By ‘strains’ of tau, we meant to refer to distinct *conformers* of tau, which are influenced by the locations of phosphorylation of tau (among many other factors). Our point, generally, was that the biochemical properties of tau are related to clinical phenotype, either between diagnoses (e.g., [Palmqvist 2020, *JAMA*]) or within an AD diagnosis (e.g., [Dujardin 2020, *Nature Medicine*]). However, to alleviate confusion on this point, we have added the modifier ‘conformational’ wherever the term ‘strain’ is mentioned throughout the text.

Comment 2.14

(Intro/Discussion) As pointed out, this paper has many limitations and makes many assumptions. There is no limitations section enumerating these, which could lead to misunderstanding by uninitiated or less expert readers.

We have added a **Limitations** subsection to the **Discussion (3.3)**; see above comments for more specific points addressed within this subsection.

Minor Comments:

Comment 2.15

It would help if Table S3 also listed how tau is quantified in each case (both the staining method and the method of quantification — the one actually shown seems to be an ordinal scoring system), as well as the time between injection (or birth in the case of non-injection) and final timepoint used in the analyses.

We have modified **Table S3** accordingly.

Comment 2.16

In section 2.5, how were the genes chosen? It is not listed in the methods, and many don’t have any known direct relationships with tau.

The AD risk genes came from the Alzheimer’s Disease Sequencing Project (ADSP) [Bellenguez 2022, *Nature Genetics*; Kunkle 2019, *Nature Genetics*]. This was mentioned in the caption for **Table S5**, but we neglected to include these details in text. We have added subsection **AD gene selection** to **Methods (5.1.3)** accordingly.

Comment 2.17

*Section 2.4 should also say “are not determined by gene expression *OR spatial information”, since neither embedding showed strong relationships with tau pathology.*

This is a valuable point, and we decided to look deeper into this question. **Figure 3** now shows two additional panels (see below): associations between mean tau pathology and the CA1-ProS cell type’s gene expression (**3B**) and spatial pattern (**3F**). The exact methodology is now detailed in **Methods (5.5)**. This further probes whether or not gene expression or spatial similarity contribute to regional vulnerability to tau.

We found that gene expression distance from CA1-ProS does not correlate with mean tau pathology, similar to **3C** and **3D**. However, we did find a highly significant association between spatial similarity to CA1-ProS and tau; this suggests that there may be a ‘vulnerability signature’ in space, which is consistent with the modest correlation in **3H** for v_3 of the spatial Laplacian:

Comment 2.18

Line 504: A brief explanation would be useful to summarize how these subclasses were chosen

These subclasses were not determined by our group, but by the authors of the Yao, et al. study. In brief, they clustered cell types at three taxonomic levels: *classes*, which differentiate types very broadly (e.g., excitatory from inhibitory neurons); *subclasses*, which contain splits within excitatory and inhibitory neurons, such as layer-specific glutamatergic neurons; and *clusters*, which contain further splits therein. In all, there were 4 classes, 42 subclasses, and 387 clusters; as described in the **Methods (5.1.1)**, we averaged across clusters to obtain subclass-specific gene expression profiles.

We have added the following language to **Methods** to clarify how these were obtained:

The scRNAseq data used to generate the cell-type maps come from Yao, *et al.* for the Allen Institute for Brain Science (AIBS), which sequenced approximately 1.3 million individual cells sampled comprehensively throughout the neocortex and hippocampal formation at 10x sequencing depth [Yao 2021, *Cell*]. Using a standard Jaccard-Louvain clustering algorithm, the authors jointly and hierarchically clustered these samples at three taxonomic levels: class ($n = 4$), subclass ($n = 42$), and cluster ($n = 387$). The full annotation and gene expression profile of each

sample, as well as trimmed mean expression across cell-type clusters, are public (<https://portal.brain-map.org/atlas-and-data/rnaseq/mouse-whole-cortex-and-hippocampus-10x>).

Reviewer #1 (Remarks to the Author):

The authors have adequately addressed all concerns. No further comment.

Reviewer #2 (Remarks to the Author):

The authors have responded to the Reviewer comments admirably. I remain enthusiastic about the strengths of this paper. The addition of a limitations section, and far more measured and accurate reporting have also substantially improved the paper's suitability for publication. Meanwhile, the additions suggested by the Reviewers have improved the paper's quality. I have now just a few unresolved issues (relating directly to previous comments) that, despite the paper's clear improvement, are important issues that remain to be addressed.

Comment 2.1

The manuscript text now does a much better job of reflecting the actual data therein, and the null modeling showcases the modest effect sizes. The authors do a very good job of discussing this. However, there remain a few prominent overstatements in the most important places: the title and the abstract.

In the abstract, the authors write:

"we have demonstrated that regional cell-type composition is a compelling explanation for the selective vulnerability observed in tauopathic diseases at a whole-brain level".

The data do not support cell types as an "explanation" of selective vulnerability. Please adjust the abstract text to reflect the fact that, while there are some associations between cell type and tau vulnerability, those associations are modest.

This is a fair point and we have revised the abstract as follows:

Neurodegenerative diseases such as Alzheimer's disease exhibit pathological changes in the brain that proceed in a stereotyped and regionally specific fashion. However, the cellular underpinnings of regional vulnerability are poorly understood, in part because whole-brain maps of a comprehensive collection of cell types have been inaccessible. Here, we deployed a recent cell-type mapping pipeline, Matrix Inversion and Subset Selection (MISS), to determine the brain-wide distributions of pan-hippocampal and neocortical cells in the mouse, and then used these maps to identify general principles of cell-type-based selective vulnerability in PS19 mouse models. We found that hippocampal glutamatergic neurons as a whole were significantly positively associated with regional tau deposition, suggesting vulnerability, while cortical glutamatergic and GABAergic neurons were negatively associated. We also identified oligodendrocytes as the single-most strongly negatively associated cell type. Further, cell-type distributions were more predictive of end-time-point tau pathology than AD-risk-gene expression. Using gene ontology analysis, we found that the

genes that are directly correlated to tau pathology are functionally distinct from those that constitutively embody the vulnerable cells. In short, we have ~~demonstrated that regional cell-type composition is a compelling explanation for the selective vulnerability observed in tauopathic diseases at a whole-brain level~~ elucidated cell-type correlates of tau deposition across mouse models of tauopathy, advancing our understanding of selective cellular vulnerability at a whole-brain level.

Similarly, and as Reviewer 1 pointed out, the title makes an overstatement, as these modest associations are hardly “underpinnings” of tau vulnerability (at least according to this data). There is also no indication of the fact that these results are all in mouse models, which the authors also admit produce results different from those published in humans. I would recommend something like:

Cellular associations with selective vulnerability to tauopathic insults in Alzheimer’s disease mouse models

or

Searching for cellular underpinnings of the selective vulnerability to tauopathic insults in Alzheimer’s disease mouse models

or something similar.

We quite like the second suggestion, and have therefore modified the title of our manuscript to be “**Searching for** cellular underpinnings of the selective vulnerability to tauopathic insults in Alzheimer’s disease mouse models”.

Comment 2.2

*In recommending the authors cross-validate their models, I previously mentioned two valid approaches — a leave-one-dataset out approach, or a k-fold cross-validation. The authors instead choose a leave-one-observation-out approach, which is not valid. This approach is no longer used in the scientific community, as it is consistently found to be optimistically biased (see e.g. Varoquaux et al. 2017 Neuroimage; Poldrack et al. 2019 JAMA Psychiatry), which is the same problem the authors started with to begin with. If the authors wish to obtain realistic estimates of how much a linear combination of cell-types can *actually* explain tau patterns, please use one of the suggested approaches.*

We apologize for the confusion as we misunderstood the reviewer’s comment. We have redone this analysis using 10-fold cross-validation on regions rather than leave-one-out. The results are qualitatively similar (see below). The LOO figures (**Figures S14** and **S18**) and table (**Table S8**) have been updated accordingly, as has the text in the **Results** describing this analysis:

1. We also performed ~~leave-one-out (LOO)~~ 10-fold cross-validation on the BIC-selected cell types per dataset to more robustly assess how well the linear models fit the data in the context of overfitting. While performance degraded as expected, all but ~~1 dataset (DS9)~~ 3 datasets retained significance at a $p < 0.01$ level (Figure S14; Table S8).
2. When we repeated the ~~LOO~~ 10-fold analysis described above (Figure S14), AD genes still generally performed worse (though less markedly so) than cell types. ~~, with the AD-gene exhibiting fits that did not reach the $p < 0.01$ significance threshold for 3 out of the 12 datasets (1 for cell types) and having higher BIC values for 8 out of the 12 datasets.~~ The AD-gene models had higher (worse) BIC values for 7 out of the 11 datasets for which at least one of the two models reached the $p < 0.01$ significance threshold, although they reached this threshold for one more dataset (10 of 12) than the cell-type models (9 of 12) (Figure S18; Table S8).

Cell types, 10-fold (Figure S14):

Genes, 10-fold (Figure S18):

Comment 2.3

The null modeling is an excellent addition to the paper and gives a much more informative estimate of effect sizes, also strengthening the author's findings. However, they still do not address the question of whether these cell type distributions are really explaining tau patterns more than anything with a similar spatial distribution (as both reviewers brought up). In the rebuttal, the authors explained that there are no such spatially-aware null models for mouse data, but techniques like Brainsmash can work on arbitrary volumes using just coordinates and values. This should be sufficient for the AMBA mapped data. This analysis is critical to understand whether the authors are learning cell-type relationships to tau, or are just learning the regional distribution of tau, which shares a lot of information but is not interesting. It would help to provide null models for Figure S9 as well.

This was an oversight on our part, and BrainSMASH was indeed suitable for this purpose. We have replaced the naive spatial null models with the autocorrelation-preserving null models generated by this tool in both **Figure 2A** and **Figure S9** (see below). The cell types identified

were largely consistent between each other, and with the naive spatial null models from the previous version. The text in the **Results** has also been modified:

To more thoroughly assess the statistical significance of each cell type's association with tau pathology across datasets, we generated distributions of mean R values using **randomly permuted spin null models** of cell-type densities **generated using the BrainSMASH toolbox [Burt 2020]** (**Figure 2A**, bottom panel; see also **Methods**). **7/8** of the 42 types exhibited significant associations ($p < 0.05$, ~~Bonferroni-corrected~~ [**Author note: Bonferroni should not be relevant for this permutation-test-based analysis; incorrectly stated before**]): CA1-ProS, CA3, **DG**, Meis2, Micro-PVM, Oligo, SUB-ProS, and Sst-Chodl (asterisks).

Pearson (Figure 2A):

Spearman (Figure S9):